# An ultrawide-range photochromic molecular fluorescence emitter

Xiao Chen[1], Xiao-Fang Hou[1], Xu-Man Chen [1] ✉ & Quan Li [1,2] ✉

Photocontrollable luminescent molecular switches capable of changing emitting color have been regarded as the ideal integration between intelligent and luminescent materials. A remaining challenge is to combine good luminescence properties with wide range of wavelength transformation, especially when confined in a single molecular system that forms well-defined nanostructures. Here, we report a π-expanded photochromic molecular photoswitch, which allows for the comprehensive achievements including wide emission wavelength variation (240 nm wide, 400–640 nm), high photoisomerization extent (95%), and pure emission color (<100 nm of full width at half maximum). We take the advantageous mechanism of modulating self-assembly and intramolecular charge transfer in the synthesis and construction, and further realize the full color emission by simple photocontrol. Based on this, both photoactivated anti-counterfeiting function and self-erasing photowriting films are achieved of fluorescence. This work will provide insight into the design of intelligent optical materials.

Artificial intelligent materials are outstanding accomplishments made by humans to imitate how living organisms respond to external stimuli, especially optical materials that can be used to create colorful, ever-changing real-time displays[1–3]. They usually cost a continuous supply of energy to maintain their advanced functions, while the light is a readily available source[4,5]. Since stimuli-response and luminescence are two critical processes that require energy for the operation of intelligent optical materials, employing light energy to power both the processes could take full advantages of light including cleanliness, noninvasiveness, remote control, spatial precision, and on-demand regulation[6–10]. Moreover, as nanoscale entities, molecular photoswitches are fundamentally important, so designing and developing molecular photoswitches that possess both excellent photochromism and photoluminescence properties becomes a promising approach but is a critical challenge in the field of chemistry and materials science[7,11,12].

A key issue is how to optimize the luminophore and the switch structure to enable a balanced utilization of light for photo-response and photoluminescence. Many organic chromophores are studied for molecular photoswitches, including azobenzenes, stilbenes, spiropyrans, and diarylethenes, whose transformation properties and functionalization are highly focused, however, they do not do

well at modulating polychromatic systems because of the lack of integration between the wavelength of response and absorption/photoluminescence[13–22]. In other words, molecular structures based on photoluminescence structure and photoswitch are usually incompatible within the same chromophore. Other approaches, such as connecting separate luminophore and photoswitchable chromophore in a non-conjugated or even non-covalent form usually results in undesired residual peaks leading to heterochromatic emission[18,23–27]. Additionally, compositing or doping molecular photoswitches into photonic crystal systems could drive luminescence shift more thoroughly, but such systems are hard to be carried out in nanoscale or in well-defined nanostructures[28–32]. Therefore, to optimize conjugated chromophores for integrated photoswitch and photoluminescence is an effective pathway for intelligent photochromic molecular light emitter.

Here, we describe a synergistic intramolecular charge transfer (ICT) and self-assembly strategy in constructing photochromic molecular light emitter to enable three advancements in functional optical materials[33–37]. These advancements are (i) single chromophore bearing both photoisomerization and photochromic luminescence with high configurational photoisomerization extent (95%), (ii) wide photochromic luminescence wavelength variation between deep blue and

[1]Institute of Advanced Materials and School of Chemistry and Chemical Engineering, Southeast University, Nanjing 211189, China. [2]Materials Science Graduate Program, Kent State University, Kent, OH 44242, USA. ✉e-mail: chenxm@seu.edu.cn; quanli3273@gmail.com

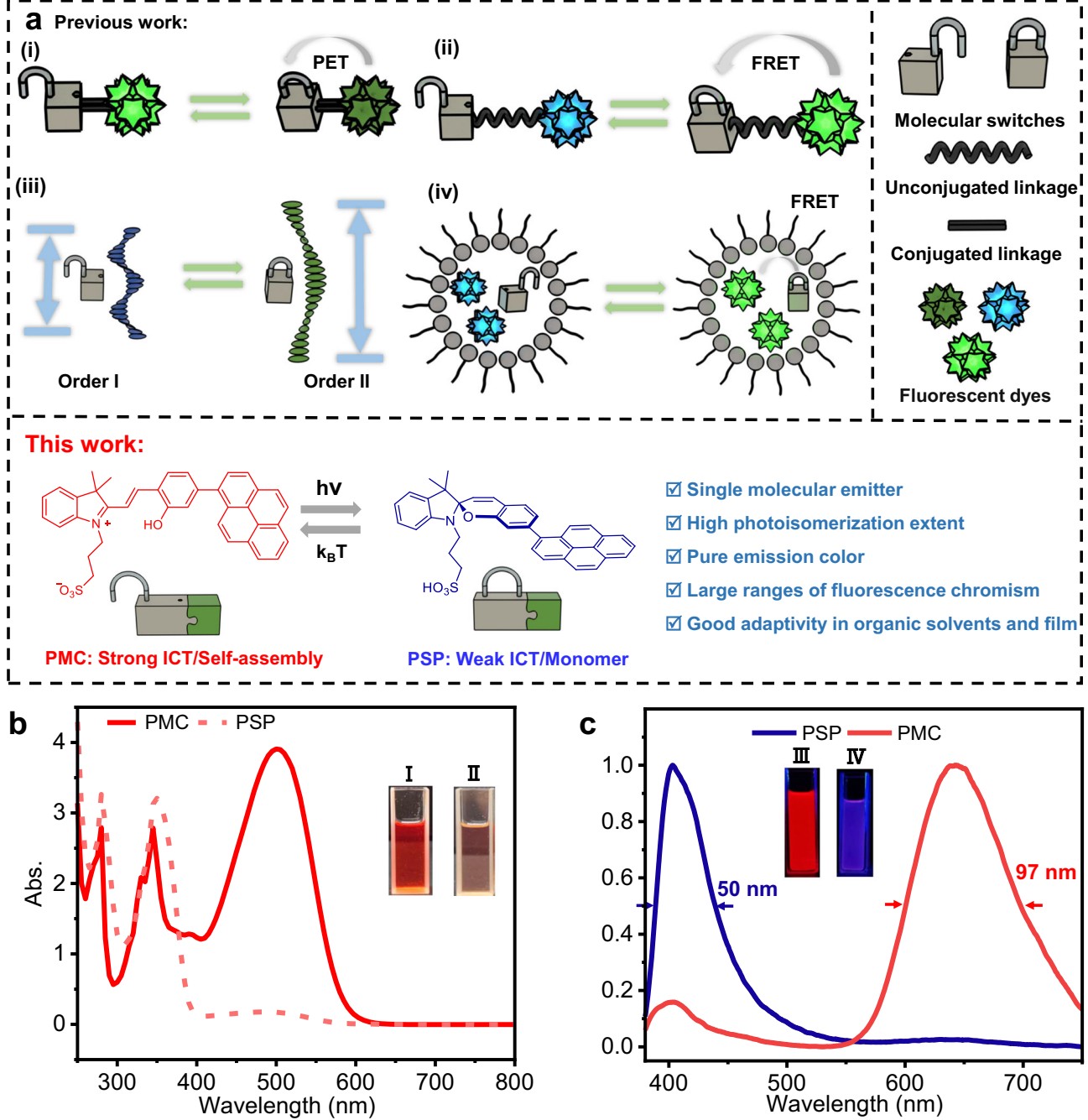

**Fig. 1 | The design strategy of ultrawide-range photochromic molecular light emitter. a** The reported strategies for photocontrollable luminescence, and pyrene-modified merocyanine-based photoswitch with wide range of photochromic fluorescence by controlling ICT and self-assembly effect in this work. (ICT: intramolecular charge transfer, PET: photoinduced electron transfer, FRET: Förster resonance energy transfer) **b**, **c** UV-Vis, and fluorescence spectra of photoswitch (0.1 mM) in strong-ICT-Self-assembly (PMC) and weak-ICT-monomer (PSP) state in CHCl₃. Inserts are image of PMC (I) and PSP (II) in natural light and PMC (III) and PSP (IV) in UV light.

red (400–640 nm), (iii) pure emission color (full width at half-maximum (FWHM): <100 nm), all of which indicate good integration of molecular switch and photoluminescence.

Merocyanine-based chromophores are highly attractive due to their reversible light-induced configurational transformation with the spiropyran form, resulting in a significant switchable emission between luminescent merocyanines and non-luminescent spiropyrans based on the controllable ICT (Fig. 1)[38–41]. However, they cannot directly lead to photochromic luminescence unless introducing external chromophores with different emission color. To overcome such restrictions, we considered if they could synergize another powerful tunable luminescence mechanism, π-π stacking from aromatic groups after self-assembling in a π-conjugated manner to achieve extreme color conversion of photochromic luminescence within a single π-expanded chromophore. In this work, pyrene, a typical moiety with assembly-controlled fluorescence emission, is directly installed on sulfonato-merocyanine into a π-expanded conjugate (PMC), both bearing light-controllable ICT and self-assembly behavior. PMC can easily photoisomerize into its spiropyran form (PSP) under light (such as 550 nm) irradiation, which further spontaneously relaxed back to PMC in the dark. Owing to the great difference in the polarity between PMC and PSP, their aggregation nanostructure

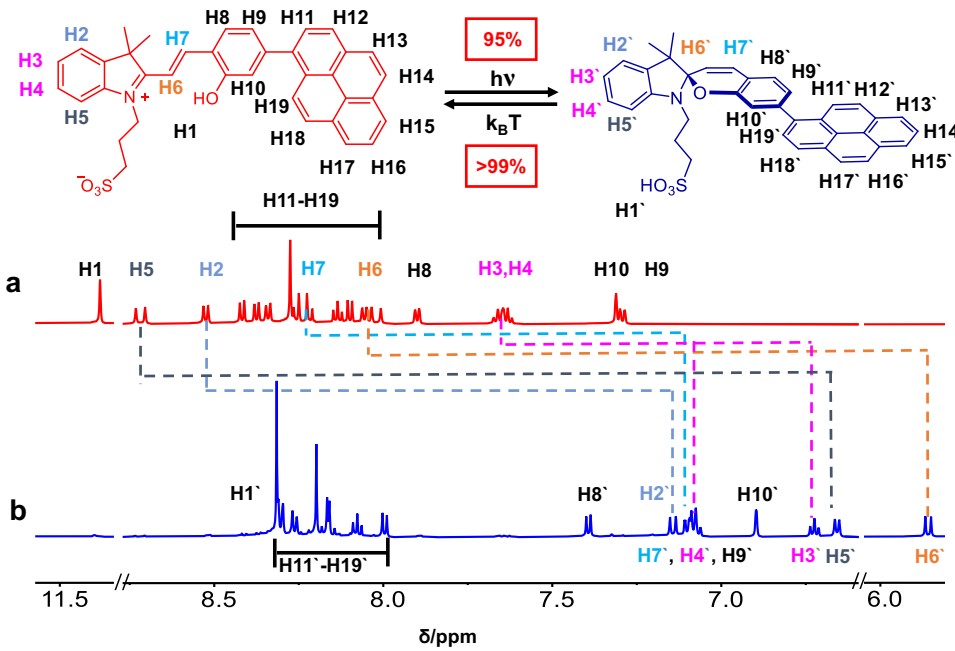

**Fig. 2 | Chemical structures and ¹H-NMR spectra (in DMSO-$d_6$) of PMC and PSP. a** Partial ¹H-NMR spectrum of PMC. **b** Partial ¹H-NMR spectrum of PSP. The variations of chemical shift of protons are marked with different color dashed lines.

and dissociation state can be easily controlled by light. Fortunately, in low-polar solvents, an ultrawide range of photochromic fluorescence between red (640 nm) and deep blue (400 nm) is realized through light-control between strong-ICT-Self-assembly state of aggregated PMC and weak-ICT-monomer state of dissociated PSP, while the range of photochromic fluorescence can be further modulated by varying the solvent polarity. According to this mechanism, the function of anti-counterfeiting of multicolor fluorescent patterns is accomplished by using a series of PMC solution of gradient polarity. Furthermore, self-erasing photowriten fluorescence pattern is also realized by doping PMC into polymer films.

## Results

### Characterization of PMC and PSP

PMC is synthesized (Supplementary Figs. 1) and characterized by ¹H-NMR,¹³C-NMR, and high-resolution mass spectra (Supplementary Figs. 27–29). Two states of the photoswitch, PMC, and PSP, are determined by ¹H-NMR, UV-Vis, and fluorescence spectra[42]. The UV-Vis absorption spectra of PMC and PSP show a huge change at ~500 nm (Fig. 1b). The ¹H-NMR shows clear peaks of PSP with negligible byproduct upon adding triethylamine, indicating PSP can be easily obtained by treating PMC with excess organic base (Supplementary Figs. 2 and 31). Thus, the photoisomerization extent (PMC to PSP) is calculated ~95%. Additionally, two sharp peaks at 330 and 345 nm change into one broad peak at 350 nm. PMC solution shows red fluorescence with a maximum emission peak at ~640 nm, however, after light irradiation, the generated PSP solution shows deep blue fluorescence, whose maximum emission peak dramatically blue-shift to ~400 nm (Fig. 1c). The FWHM of PMC and PSP fluorescence peaks are calculated as 97 nm and 50 nm, respectively, which demonstrate that both the two states of such photoswitch meet the high purity of emission color among the organic light emitters[43,44]. Emission quantum yield and fluorescence lifetime of PMC and PSP are 5.12%, 1.71 ns, and 3.35%, 0.28 ns (Supplementary Fig. 3). Furthermore, such photo-switch is responsive to a wide range of wavelength from 475 to 600 nm in high photoisomerization extent (>95%) (Supplementary Fig. 4 and Supplementary Table 1). Quantum yield of photoisomerization for PMC in CHCl₃ with different light is also calculated (Supplementary

Table 1). The quantum yield of photoisomerization is 0.24 when the light source is white light and the quantum yield reach 0.64 while the light wavelength is 475 nm. The phototransformation from PMC to PSP state is also confirmed by ¹H-NMR (Fig. 2, Supplementary Figs. 27 and 30). With the molecular skeleton changing from mer-ocyanine to spiropyran, all the protons shift to high-field area. The proton of phenolic hydroxyl group at 11.3 ppm disappears, while the H of sulfonic acid group emerges at 8.32 ppm. The peaks of protons near N atom (H2–H7) also show large shifts upon the change of charge at N atom. The ¹H-NMR results demonstrate that the photoisomerization from PMC to PSP has high transformation efficiency and little side photoreactions, which is consistent with UV-Vis and fluorescent spectra results.

To further study the mechanism of tunable ICT-Self-assembly effect inducing photochromic fluorescence, we proposed the iso-merization of the photoswitch (Fig. 3a, b)[40,41,45,46]. PMC is a zwitterionic merocyanine structure bearing *trans* C=C double bond, while PSP is a spiropyran sulfonic acid bearing a six membered ring with a *cis* C=C double bond. Hence, there are two processes in the isomerization from PMC to PSP, including *cis-trans* isomerization and cyclization. PMC is predominant and stable in the dark, while it obtains light energy to form PMC (S1), which is 70.7 kcal/mol above the ground state. PMC rapidly relaxes to a minimum on the S1 potential energy surface, which is PMC′ (S1) with the C1–C2 bond bears a vertical geometry, resulting in the *trans-cis* isomerization once returned to the ground state to form intermediate 1. Then, intermediate 1 readily undergoes a nucleophilic cyclization through TS1, with a negligible barrier of 3.1 kcal/mol. In this process, the −OH group acts as an intramolecular nucleophile, and the proton is transformed by the −SO₃⁻ group in a concerted manner, finally releasing the product PSP, which is 3.2 kcal/mol higher in free energy than PMC. Therefore, PSP could only be enriched under the photo-equilibrium condition. PSP thermally relax back to PMC in the dark to finish the cycle of reversibly light-driven and thermal relaxation process, which achieved by the retro-cyclization through TS1, and the subsequent ground state bond rotation process. The C1−C2 bond rotation relies on the donor ability of the phenol moiety towards the imine cation fragment, which reduces the double bond character of C1−C2, enabling a ground state cis-trans

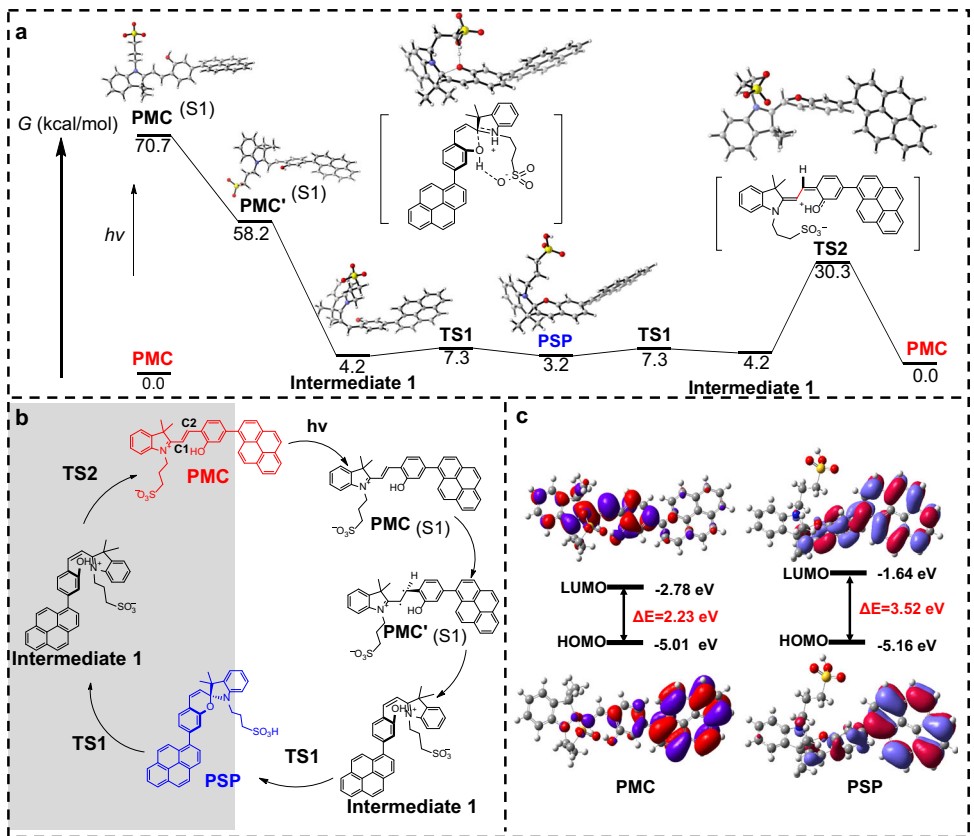

**Fig. 3 | Proposed mechanism of photoisomerization/thermal relaxation process and HOMO/LUMO simulation of PMC and PSP. a, b** Gibbs free energy profile (kcal/mol) and schematic representation of the photoswitch isomerization mechanism (obtained by M06-2X/6-311 + G(d,p)//M06-2X/6-31 + G(d,p)). **c** The frontier molecular orbitals of PMC and PSP (optimized by B3LYP/6-31 + G). (TS: transition state, G: Gibbs free energy).

isomerization with a barrier of 26.1 kcal/mol, and finally leads to the formation of the thermodynamically favored PMC. Moreover, for insights into the relationship of the structural properties between PMC and PSP, quantum calculations are performed on evaluating the lowest unoccupied molecular orbital (LUMO) to highest occupied molecular orbital (HOMO) through density functional theory (DFT). As shown in Fig. 3c, the HOMO of PMC mainly delocalizes at indole and sulfonic acid moieties, while its LUMO delocalizes at the whole area of the molecule, indicating a typical ICT property of PMC to reduce the bandgap between HOMO and LUMO. In contrast, both the HOMO and LUMO of PSP mainly delocalize at the pyrene moiety, indicating a typical and high HOMO-LUMO bandgap. The HOMO-LUMO energy difference (ΔE) of PMC and PSP are calculated as 2.23 eV and 3.52 eV, respectively, which is fortunately consistent with their experimental UV-Vis absorption and further confirms the hypochromic shift from PMC to PSP on both UV-Vis absorption and fluorescence emission. These results show good agreement with the experimental results and strongly supports the controllable ICT effect to realize a wide range of photochromic fluorescence.

## Self-assembly behavior of PMC

The self-assembly behavior of PMC is investigated to demonstrate that contributes to enlarging the wavelength range of photochromic fluorescence. A series of PMC solution in CHCl₃ in gradient concentrations (0.0001–0.1 mM) is analyzed by UV-Vis absorption and fluorescence emission. The absorbance at 500 nm of PMC shows a linear variation as well as no obvious shifting of UV-Vis spectra upon the concentration change of PMC solution, indicating that PMC do not exhibit any intermolecular complexation in the ground state (Fig. 4a and Supplementary Fig. 5a). In contrast, PMC shows different performance of fluorescence when varying its concentration in CHCl₃. At a

trace concentration (0.0001 mM), PMC exhibits blue fluorescence (~400 nm), while the fluorescence color changes into red (~640 nm) when the concentration increases up to 0.1 mM (Fig. 4b). Meanwhile, the life decay for the PMC in CHCl₃ (0.03 mM) shows that the emission at 640 nm of PMC has a noticeable rise time of ca. 2.2 ns, while the rise time of emission at 400 nm is ca. 1.6 ns. (Supplementary Fig. 5b). The longer rise time (~1.4 times) for the emission at 640 nm of PMC might result from some π-π stacking structure (like excimer) forming in self-assembly, which in accordance with the time-resolved fluorescence (TRF) spectra (Supplementary Fig. 5c) for the PMC in CHCl₃[47,48]. As the concentration of PMC solution increases from 0.0001 to 0.1 mM, the optical transmittance at 625 nm decreases in two different linear trends, revealing a critical assembly concentration of ~0.039 mM upon linear fitting, where obvious Tyndall effect appears in 0.1 mM PMC in CHCl₃ (Fig. 4c and Supplementary Fig. 6). Interestingly, the scanning electron microscopy (SEM) image shows several doughnut-like self-assembled structures with average ~2 μm diameter (Fig. 4d). We speculate that the shown morphology in SEM images attributes to both the self-assembly behavior in the original PMC solution in CHCl₃ and the further aggregation process of PMC during drying the sample for SEM imaging. The energy distribution spectra (EDS) results show that the key elements content of these doughnut-like self-assembled structures are highly consistent with PMC (S: 3.53 wt.%, N: 2.07 wt.%), demonstrating PMC as the main component of the building block (Supplementary Fig. 7). Therefore, based on the controllable synergistic self-assembly and ICT effect, the ratio of fluorescence intensity of PMC at 400 nm and 640 nm significantly decreases when increasing the concentration, showing a series of fluorescence colors including blue, purple, pink, and red (Fig. 4e–g). On the contrary, PSP only shows some stacked and irregular nanostructures attribute to the precipitation during drying the SEM sample, indicating that no self-assembled

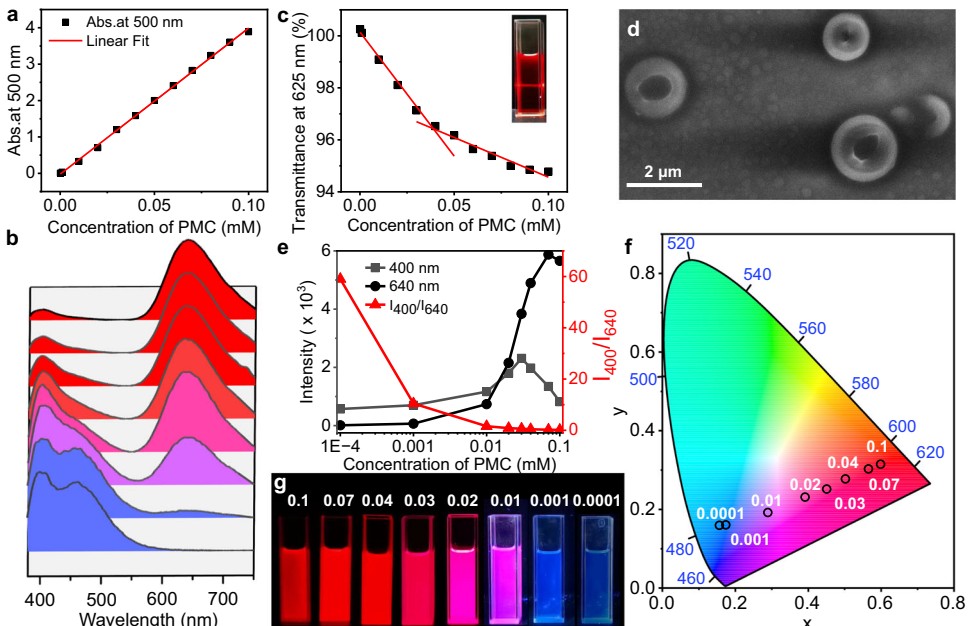

**Fig. 4 | Investigation for the ICT-Self-assembly mechanism enabling photo-chromic fluorescence of PMC in CHCl₃ solution. a** Absorption at 500 nm of PMC from 0.0001 mM to 0.1 mM. **b** Normalized fluorescence spectra of PMC with concentration from 0.1 mM, 0.07 mM, 0.04 mM, 0.03 mM, 0.02 mM, 0.01 mM, 0.001 mM, and 0.0001 mM in chloroform (from top to bottom, $\lambda_{ex}$ = 365 nm). **c** Optical transmittance at 625 nm of PMC from 0.0001 mM to 0.1 mM. **d** SEM image of PMC (0.1 mM). **e** The 640 nm emission, 400 nm emission, and the ratio of the two emission intensities ($I_{400}/I_{640}$) of PMC in different concentration. **f, g** The CIE 1931 chromaticity diagram, and fluorescence images of PMC in different concentration. ($\lambda_{ex}$ = 365 nm).

structures form in PSP state in CHCl₃ (Supplementary Fig. 8). Both UV-Vis absorption and fluorescence emission of PSP show negligible shift when varying its concentration, indicating dissociated PSP in CHCl₃ cannot form self-assembly (Supplementary Figs. 9 and 10). Therefore, the ICT-Self-assembly synergistic mechanism is established to enable a wide range of photochromic fluorescence from blue to red.

## Dynamics of photoisomerization and thermal relaxation processes

Because light irradiation and thermal relaxation take responsibility for the reversible transformation between PMC and PSP, the dynamic processes of the photoswitch are further investigated. Upon white light irradiation, the absorbance of PMC at 500 nm rapidly decreases in ~10 s to form PSP and remains unchanged for another ~110 s, and then gradually relaxes back to PMC in the dark for ~300 min (Fig. 5a and Supplementary Fig. 11). At least 5 cycles can the PMC/PSP photo-transformation be executed with negligible changes, indicating that such photoswitch exhibits excellent reversibility. We fitted the variation of absorbance at 500 nm of five light irradiation and thermal relaxation processes to find that the two processes operate according to first-order reaction kinetics. (Supplementary Figs. 12 and 13 and Supplementary Tables 3 and 4). The half-lives of light irradiation and thermal relaxation processes undergoes little changes in different cycles, and the average half-lives of the two processes are calculated (2.40 ± 0.24) s and (2000 ± 52) s, respectively (Fig. 5c and Supplementary Fig. 12). The photoisomerization extent of light irradiation and thermal relaxation processes are (94 ± 0.007)% and (100 ± 0.03)%, respectively, which is considered as a stable and nearly complete transformation in several cycles. Furthermore, upon light irradiation, the fluorescence emission at 640 nm and 400 nm sharply decreases and increases within ~10 s and reaches unchanged after another ~110 s, respectively, which is consistent with the UV-Vis dynamics results (Fig. 5b and Supplementary Fig. 14). Similarly, the thermal relaxation process also requires ~300 min back to the emission of the initial PMC solution. The photochromic fluorescence of PMC/PSP system also

shows good reversibility in 5 cycles without any decline. The half-lives of light irradiation and thermal relaxation processes are calculated (3.68 ± 0.43) s for 640 nm, (4.10 ± 0.82) s for 400 nm, (3222 ± 95) s for 640 nm and (2731 ± 156) s for 400 nm, respectively (Fig. 5c, Supplementary Figs. 15 and 16 and Supplementary Table 5). The hysteresis of half-life calculated according to fluorescence spectra data also indicate that ICT-Self-assembly-induced fluorescence variation occurring in this process, that is the 640 nm emission resulting from self-assembly. The shorter half-life in light irradiation and longer half-life time in thermal relaxation of 640 nm emission further support this statement. The fluorescence of PMC in the latter cycles appears even a little stronger because the slightly lower concentration of PMC after thermal relaxation causes a higher emission according to the emission standard curve of PMC. Thus, taking advantage of ICT-Self-assembly-induced fluorescence variation and excellent isomerization reversibility, time-dependent polychromatic fluorescence between deep red and deep blue is achieved. Both CIE 1931 chromaticity and digital images under 365 nm excitation light show the dramatic variation of the fluorescence wavelength during the spontaneous thermal relaxation process (Fig. 5d–f). The dynamic fluorescence color changes from blue, cerulean, violet, purple, pink, and magenta, and finally become red, which is similar with that of the initial PMC solution.

## Photochromic fluorescence behavior in solvents and film

Based on the above results of photocontrollable ICT-Self-assembly system, we wonder whether different photochromic fluorescence behaviors can be realized by taking advantages of solvent effect. One feasible way is to change the polarity of solvent to control the aggregation state of PMC and PSP. PMC is more likely in aggregation state in low-polar solvents such as CHCl₃ because of its high-polar zwitterionic structure, while PSP is more likely aggregation state in high-polar solvents because PSP possess a relatively low-polar structure without any charges. Thus, various solvents are used to study the photo-controlled self-assembly behavior of PMC and PSP. Firstly, the photo-isomerization and thermal relaxation behaviors of PMC are

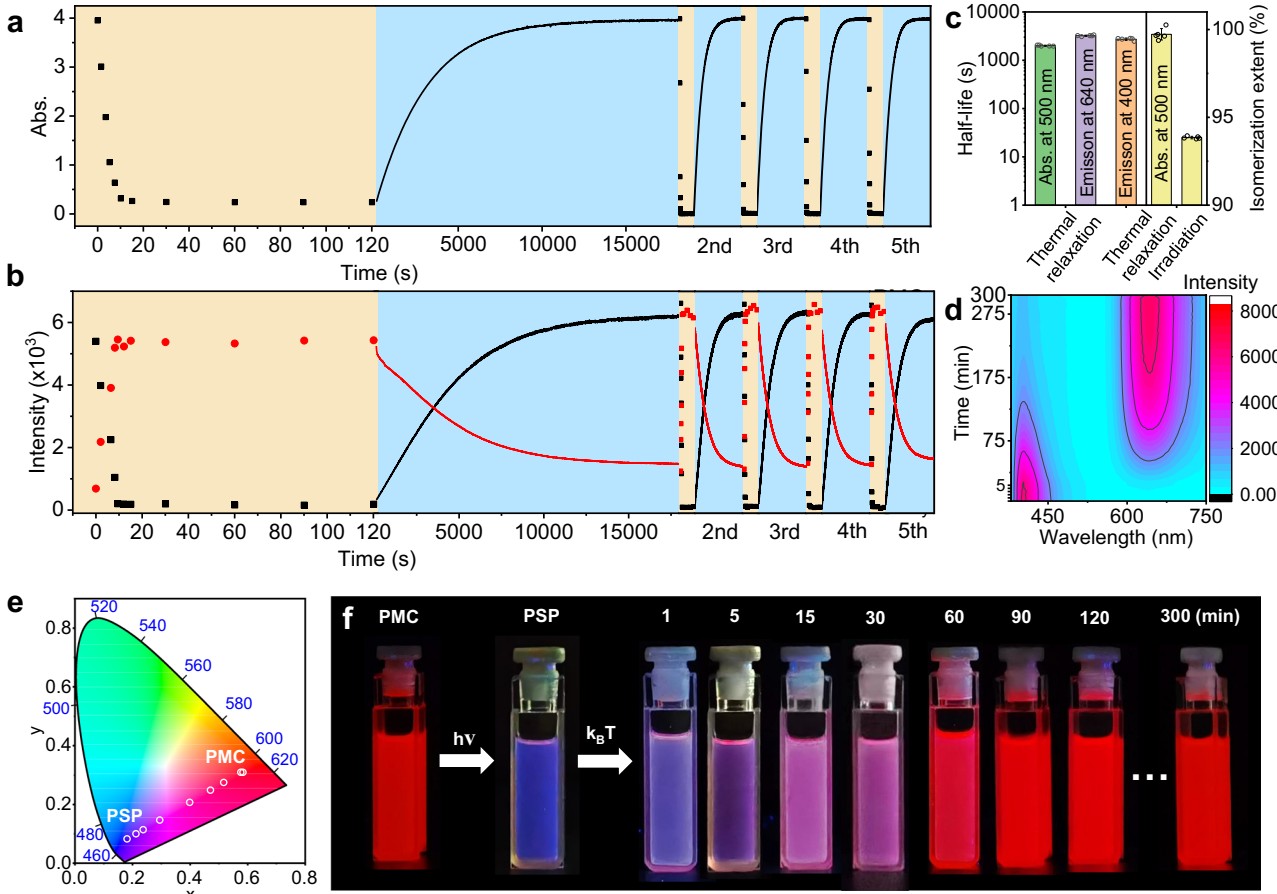

**Fig. 5 | Characterization of the dynamics of photoisomerization and thermal relaxation, and the time-dependent fluorescence chromism of PMC photoswitch in CHCl₃ (25 mW cm⁻², 25 °C unless mentioned). a, b** Absorbance at 500 nm and emission at 400 nm/640 nm for 5 cycles of time-dependent photoisomerization and thermal relaxation process. The yellow areas represent photoisomerization process, and the blue areas represents thermal relaxation process. The rate constant of photoisomerization process $k_2$ is $(0.29 \pm 0.03)$ s⁻¹, and rate constant of thermal relaxation process $k_3$ is $(223.10 \pm 5.59)$ s⁻¹ (see Supplementary Tables 3 and 4). **c** Half-life of thermal relaxation process and photoisomerization extent of light irradiation and thermal relaxation process through time-dependent dynamic UV-Vis absorption and fluorescence emission spectra. n = 5 independent experiments, with the bar data indicating mean ± SD. **d** Time-dependent 3D-fluorescence spectra of thermal relaxation process. **e, f** The CIE 1931 chromaticity diagram, and fluorescence images of thermal relaxation process at certain time including 0 min (PSP state), 1 min, 5 min, 15 min, 30 min, 60 min, 90 min, 120 min, and 300 min.

demonstrated well in all these solvents except water (Supplementary Figs. 17 and 18). The photoisomerization and thermal relaxation efficiency are all higher than 40%, and the half-lives of the thermal relaxation processes are also different, indicating PMC possesses good adaptivity to show photochromic fluorescence in various solvents (Supplementary Table 6). Moreover, PMC shows different photochromic fluorescence variation in these solvents, which mainly depends on their polarities. As shown in Fig. 6a and b, in low-polar solvents, the strong-ICT-Self-assembly state of PMC with red fluorescence isomerizes into weak-ICT-monomer state of PSP with blue fluorescence. However, in high-polar aprotic solvents, PMC is in monomer state, and the only ICT effect can make its fluorescence shift to yellow, while PSP is in aggregation state to induce a cyan emission of self-assembly state. For example, in DMSO, the absorption wavelength of PMC is a little shorter (465 nm) than that in CHCl₃, indicating a dissociation state of PMC in DMSO (Supplementary Fig. 19). Because of the dissociation state, the photoisomerization extent of PMC in DMSO is a little higher than that in CHCl₃. The broaden peaks of ¹H-NMR spectrum of PMC in CDCl₃ and DMSO-$d_6$ also indicate the aggregation state and dissociation state in CHCl₃ and DMSO solvents, respectively (Supplementary Fig. 20). The maximum fluorescence emissions of PMC and PSP in DMSO are 550 nm and 460 nm, respectively, and other high-polar aprotic solvents show the similar performance

(Supplementary Fig. 17 and 18). In high-polar protic solvents such as ethanol, the enhanced ICT effect induce emission of PMC shift to red, while PSP cannot aggregate because the ionization of sulfonic acid group lead to a higher polarity of PSP (Supplementary Fig. 21). In a word, the fluorescence colors of PMC and PSP are dependent on the states (monomer or self-assembly state) upon varying the solvents. Compared to high-polar solvent, low-polar solvent leads to self-assembly of PMC owing to the high-polar zwitterionic structure. This is beneficial for ICT-Self-assembly state, resulting in a large redshift of fluorescence. Meanwhile, the closed formed of PSP leads to a different fluorescence variation.

According to the results above, a facile strategy is generated to apply a mixture of a low-polar solvent and a high-polar aprotic solvent to control the color ranging of the photochromic fluorescence of PMC. Two representative solvents, o-dichlorobenzene (DCB) and DMSO, are chosen to demonstrate such a design strategy and exhibit the anti-counterfeiting function. The photochromic fluorescence behavior of PMC in DCB is highly consistent with that in CHCl₃, and the low volatility of DCB makes it more suitable as a solvent to perform light-induced anti-counterfeiting. When using DCB/DMSO mixed solvents in different ratios, PMC shows significant fluorescence chromism including red, orange, and yellow, while PSP shows similar cyan color of fluorescence (Fig. 7a, b, and Supplementary Fig. 22). Thus, a series of

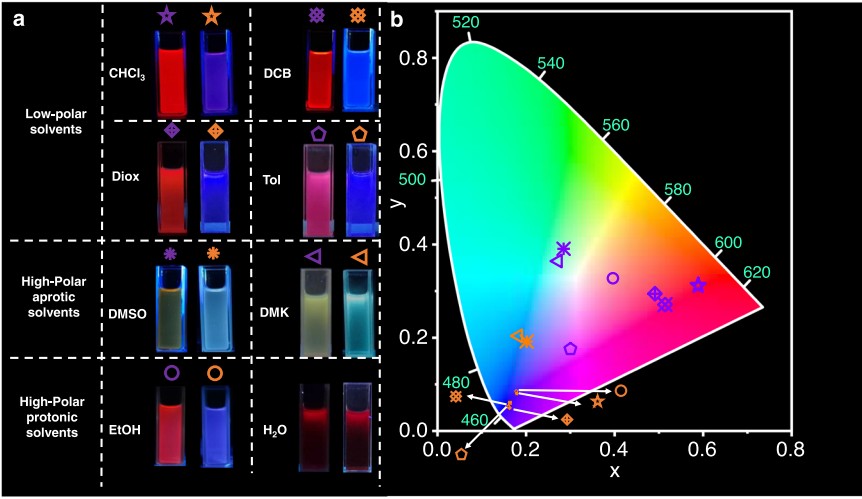

**Fig. 6 | Solvent effect of photochromic fluorescence of PMC. a, b** Fluorescence images and CIE 1931 chromaticity diagram of PMC and PSP in CHCl₃, o-dichlorobenzene (DCB), 1,4-dioxane (Diox), toluene (Tol), DMSO, acetone (DMK), ethanol (EtOH) and water. The purple mark represents PMC state, and orange marks represents PSP.

PMC solutions in different ratio of DCB/DMSO solvents are used to prepare a multicolor fluorescent SEU pattern in a porous plate (Fig. 7c). Upon 365 nm excitation light, the SEU pattern in yellow and orange fluorescence color emerges with a red fluorescence background, while all the background and pattern become cyan fluorescence under 365 nm excitation light after treating with white light irradiation (Fig. 7d). Such fluorescence pattern can further recover after leaving it in the dark for 12 h (Supplementary Fig. 22). Therefore, to demonstrate a possible future application of this dynamic photoswitch system, we developed an anti-counterfeiting proof of concept experiment by using PMC-based photochromic fluorescence system upon facile control of the ratio of mixed solvent.

Furthermore, to demonstrate more potential as an excellent molecular fluorescence emitter of PMC with a wide range of photochromic fluorescence in different systems, polymethyl methacrylate (PMMA) film is employed as another kind of substrate. The PMC-PMMA film is prepared by evaporating a PMC-PMMA solution in CHCl₃ on a glass plate. The PMC-PMMA film appears bright orange, while the film becomes light-orange after light irradiation (Supplementary Fig. 23). The UV-Vis absorption spectra show a significant decrease at 490 nm, indicating that PMC maintains a high phototransformation efficiency to PSP (Fig. 7e). The fluorescence emission also appears a significant variation from orange (601 nm) to deep blue (401 nm), and the FWHM is calculated as 46 and 90 nm, respectively (Fig. 7f). Because of over-stacked molecule in fibrous cavities of PMMA limiting the molecular motion, the photoisomerization and thermal relaxation process of PMC occurs much more slowly than that in solvent, whose half-lives are calculated to be 3.37 s and 49297 s, respectively (Supplementary Fig. 24). The photowriting and self-erasing functions of the PMC-PMMA film are further demonstrated (Fig. 7g). The film is exposed under white light with a photomask, specific image (the emblem of Southeast University) could be successfully obtained after removing the irradiation light, which is clearly under 365 nm light irradiation. Furthermore, the image can be erased easily upon full exposure under white light for two minutes and recover slowly back to the initial state in the darkness for use in the next cycle (Fig. 7h).

## Discussion

By means of both powerful structural design and simple synthesis, we have accomplished a highly controllable and adaptable molecular photochromic fluorescence emitter with excellent luminescence performance. By directly conjugating merocyanine and pyrene moieties,

comprehensive properties of photoswitch, ICT, and self-assembly play their full roles to build up a photoresponsive time-dependent multi-color luminescence system. We demonstrate the mechanism of tunable ICT-Self-assembly-induced fluorescence change to explain the ultrawide variation (640–400 nm) of the emission wavelength between the two states of the photoswitch. Solvent polarity can control the self-assembled aggregation behavior, allowing the change of the synergistic effect with ICT, resulting in different ranges of fluorescence variation. Its excellent photochromic fluorescence performance in PMMA films further demonstrates its adaptivity in different environments. Based on such design and mechanism, this work would provide a means to fabricate intelligent luminescent materials and set the foundation for next-generation light-manipulative molecular and nanostructured fluorescence devices and beyond.

## Methods

### Synthesis of (E)-3-(2-(2-hydroxy-4-(pyren-1-yl)styryl)-3,3-dimethyl-3H-indol-1-ium-1-yl)propane-1-sulfonate (PMC)

2,3,3-trimethyl-1-(3-sulfonatepropyl)-3H-indolium (100 mg, 0.36 mmol) and 2-hydroxy-4-(pyren-1-yl)benzaldehyde (1.1 eq) was added into ethanol. The reaction was heated to 85 °C for 3 days. After the reaction, the mixture was dropped into ethyl acetate. The precipitate was collected by filtration, washed with ethyl acetate and n-hexane three times to obtain 0.1 g deep red product (yield: 48.07%). $^1$H-NMR (600 MHz, DMSO-$d_6$) δ: 11.34 (s, 1H), 8.72 (d, J = 16.3 Hz, 1H), 8.52 (d, J = 8.1 Hz, 1H), 8.41 (d, J = 8.0 Hz, 1H), 8.38–8.32 (m, 2H), 8.29–8.20 (m, 4H), 8.13 (t, J = 7.6 Hz, 1H), 8.09 (d, J = 7.8 Hz, 1H), 8.07–7.97 (m, 2H), 7.90–7.87 (dd, J = 7.2, 1.2 Hz, 1H), 7.68–7.61 (m, 2H), 7.32 (d, J = 1.7 Hz, 1H), 7.29 (dd, J = 7.9, 1.9 Hz, 1H), 4.86 (t, J = 7.9 Hz, 2H), 2.69 (t, J = 6.5 Hz, 2H), 2.24 (m, 2H), 1.84 (s, 6H). $^{13}$C NMR (151 MHz, DMSO-$d_6$): δ: 182.10, 159.48, 148.76, 148.20, 143.98, 141.46, 136.21, 131.42, 131.19, 130.83, 130.60, 129.66, 129.60, 128.65, 128.39, 127.97, 127.84, 127.64, 127.08, 126.19, 125.81, 125.54, 124.80, 124.61, 124.43, 123.48, 122.88, 121.16, 118.75, 115.57, 112.06, 52.40, 47.89, 46.05, 26.96, 25.13. HRMS (m/z): [M-H]⁻ calcd. for C₃₇H₃₁NO₄S, 584.18901, found, 584.19141.

### General method of photoisomerization and thermal relaxation processes of PMC in solvents

PMC solution is filled into a quartz cell and set under a light irradiation for a period of time for photoisomerization process. The obtained solution is then set in the dark for thermal relaxation process. The light

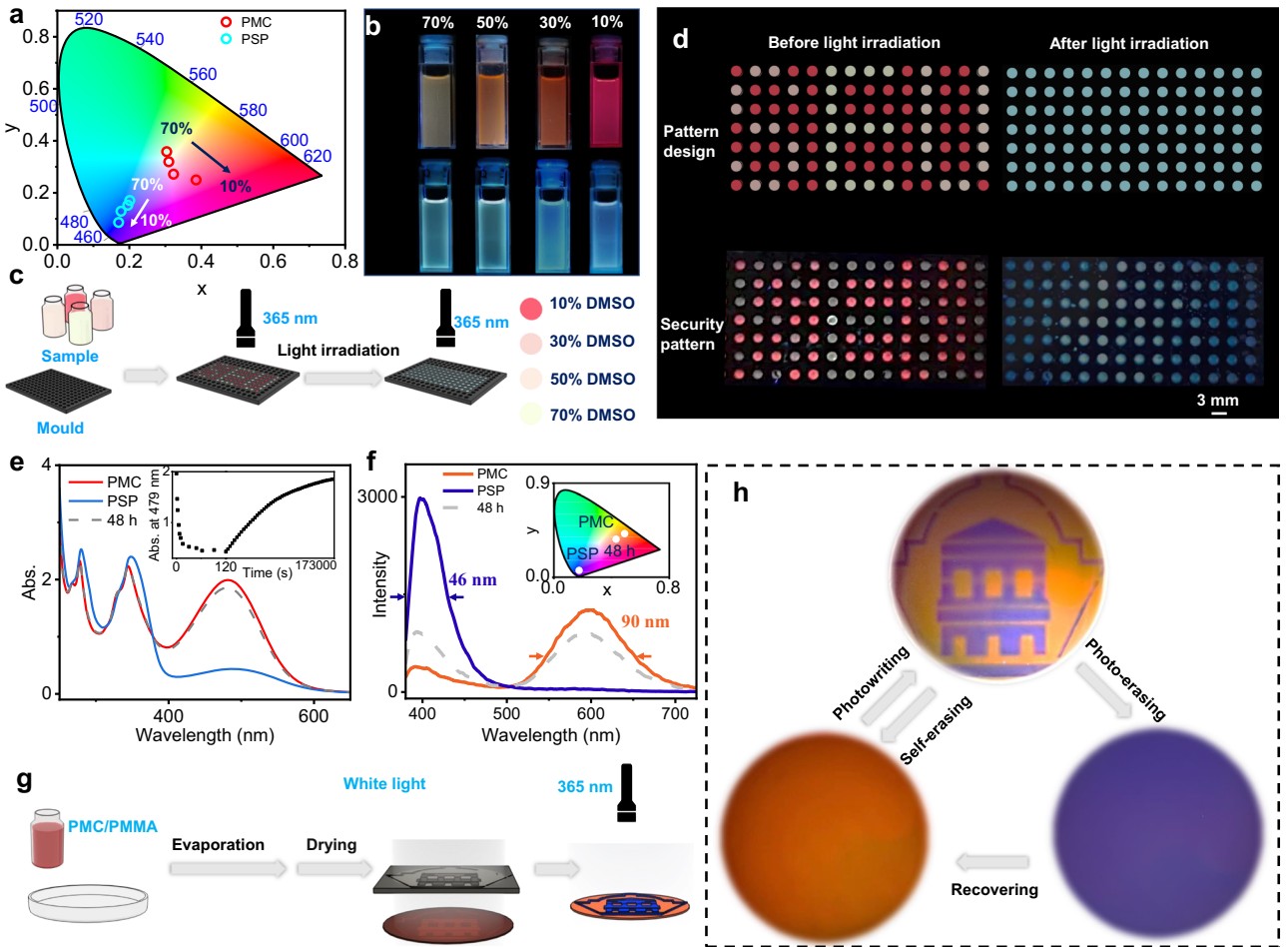

**Fig. 7 | The control of photochromic fluorescence ranges using a series of mixed solvents (DMSO/DCB) and photowriting and self-erasing behavior of PMC-PMMA film. a** The CIE 1931 chromaticity diagram of PMC and PSP in the DMSO/DCB solvent with different DMSO fractions. **b** Fluorescence image of the photoswitch in the DMSO/DCB solvent with different DMSO fractions. The top ones are PMC and bottom ones are PSP. **c**, **d** The illustration, and digital images of anti-counterfeiting patterns of PMC or PSP in mixed solvents. **e**, **f** Fluorescence and UV-Vis spectra for photoisomerization and thermal relaxation process of PMC-PMMA film. The insert spectra are thermal relaxation dynamics of absorption at 479 nm of the film and CIE 1931 diagram, respectively. **g**, **h** The illustration, and digital images for photowriting, full light-exposure, and self-erasing process of the film.

power density is 25 mW cm⁻², and the temperature is 25 °C unless mentioned. The excitation wavelength is 365 nm for the fluorescence in this work unless mentioned.

### Investigation for reaction orders of photoisomerization and thermal relaxation processes

All data of photoisomerization and thermal relaxation processes obtained from UV-Vis and fluorescence spectra at certain wavelengths are plotted and fitted with first-order kinetic equation:

$$\ln[A] - \ln[A_0] = -k_1 t \qquad (1)$$

[A] refers to [PMC] or [PSP]. [PMC] is obtained according to its UV-Vis standard curve in Fig. 4a, and the [PSP] is obtained from the difference between [PMC₀] and the remaining [PMC].

Furthermore, half-life is used to evaluate the reaction rates of the photoisomerization and thermal relaxation processes. The half-life is defined as the time that the concentration of PMC reached the average concentration of the initial and the end state in the solution with the specific experimental conditions. The half-lives for the photoisomerization process were determined by reading the time that the concentration or fluorescence emission reached the average of the

initial and the end state during the concentration or fluorescence variation.

The experiment condition: temperature is 25 °C. Clearly and transparent PMC solution (CHCl₃) with concentration is 1 × 10⁻⁴ M. The light source is a Xenon lamp (Ceaulight, CEL-HXF300) and the light power density is 25 mW cm⁻². For investigation half-life for the photoisomerization, the solution is irradiated by a certain time at each time. The total irradiation time is 2 minutes.

### Film preparation

2.5 g PMMA is dissolved in 20 mL CHCl₃ at 50 °C, and then 0.43 mL PMC solution (2.5 mg in 0.43 mL DMSO, 1 × 10⁻² M) is added. The formed PMC/PMMA solution is poured into a glass plate (Φ7.5 cm). After degassing, This PMC/PMMA solution in glass plate is transferred into dark place for evaporating solvent ~2 days to obtain film. Then keep the film in vacuum for another 1 day under 40 °C to remove the residual solvents completely. The photomask of the auditorium pattern of Southeast University is customized commercially.

### UV-Vis spectroscopy

UV-Vis spectra and the optical transmittance are recorded in a quartz cell (light path 10 mm) on a Shimadzu UV-2700 spectrophotometer equipped with a temperature controller. The UV-Vis spectra of film are

recorded with film holder, and the thickness of film was measured as 0.5 mm.

## Fluorescence spectroscopy
Steady-state fluorescence spectra are recorded in a conventional quartz cell (light path 10 mm) on a Hitachi F-4700 equipped with a temperature controller. All steady-state fluorescence spectra are recorded by normal 90° method. The excitation wavelength is 365 nm (unless mentioned). Steady-state fluorescence spectra of the film were recorded with a film holder, and the thickness of the film was measured as 0.5 mm. The emission quantum yield and fluorescence lifetime are recorded in a conventional quartz cell (light path 10 mm) on a Horiba PluoroLog 3-TCSPC. The time-resolved fluorescence spectrum is recorded by a conventional quartz cell (light path 10 mm) on Edinburgh FLS-1000.

## SEM imaging and EDS
SEM images and EDS test are obtained using a Nova Nano SEM450 scanning electron microscope equipped with EDS analysis. The sample for high-resolution SEM measurements is prepared by dropping the solution onto a silicon wafer. The sample on the wafer is then air-dried.

## NMR spectroscopy
$^1H$ and $^{13}C$ NMR spectra are recorded on a Bruker HW600 MHz (AVANCE AV-600) spectrometer.

## HRMS
The electron spray ionization mass spectrometry is measured by Thermo Scientific Q Exactive.

## Data availability
The authors declare that data supporting the findings of this study are available within the paper and its Supplementary Information, and from the authors upon request. Source data for DFT calculation are provided. Source data are provided in this paper.

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

## Acknowledgements

We thank the Jiangsu Innovation Team Program, NNSFC (22001035), the Priority Academic Program Development of Jiangsu Higher Education Institutions, and Zhishan Scholars Programs of Southeast University for financial support. We thank Hu Jun and Zhu Guanqun from Southeast University for PMMA film preparation.

## Author contributions

Q.L. and X.-M.C. directed and designed the project. X.C. performed all the experiments. X.-M.C., X.C., and X.-F.H. analyzed all the data. Q.L., X.-M.C., and X.C. co-wrote the paper. All authors discussed the results and commented on the manuscript.

## Competing interests

The authors declare no competing interests.
