## [Peer Review File · Nature Communications]

An ultrawide-range photochromic molecular fluorescence emitterREVIEWER COMMENTS

Reviewer #1 (Remarks to the Author):

The manuscript “An ultrawide-range photochromic molecular fluorescence emitter” by Quan Li, Xu-Man Chen and co-workers considers photochromic fluorescent molecules with a broad spectral range.

The essence of the paper is that the authors have created a photoswitch system that combine ICT state engineering with controlled excimer formation to create new advanced functional materials. In general the work is interesting and well carried out, my comments are mainly to the language and the use of specific terms such as “intelligent materials” which I personally think should be clarified.

All together, I think that this is a very nice piece of work that deserves publication in Nature Commun, in my view, it has all the needed elements: it has a novel physical effect and very nice experimental verification. I however have quite a number of minor comments that I suggest that the authors look into, these comments are given with the best intention as feedback on possible ways on how to improve the manuscript further.

1) The authors write “Here, we describe a synergistic intramolecular charge transfer (ICT) and excimer strategy in constructing photochromic molecular light emitter to enable three advancements in intelligent optical materials”

I need to be convinced on the use of the term “intelligent” here I would prefer the term “functional”

2) figure 3 proposes a mechanism and calculates orbitals and energy levels of some of the involved species. To verify the proposed mechanism, I suggest to calculate the energies of all proposed intermediates and transition states and construct an energy-reaction coordinate plot. The terms T1-T3 are a bit difficult to understand here, are they transition states? Or transient species? By definition, you cannot have a conversion from a transition state to a transition state. Maybe what the authors mean is that T1-3 are intermediates? I suggest to remake figure 3 with correct TS and intermediate structures, and then revise the discussion accordingly. If the proposed mechanism is correct, there should be 3 intermediates, two photo isomers and 5 transition states. Or? Please clarify.

3) the authors uses the term “The isomerization efficiency” is that the “photostationary state” ? what is the quantum efficiency of conversion? Please clarify.

4) the sentence “The half-lives of light irradiation and thermal relaxation processes undergoes little changes in different cycles, and the average half-lives of the two processes are calculated (2.40 ± 0.24) s and (2000 ± 52) s” is very difficult to understand. In normal photoswitch systems, there can only be one thermal half life? The rates and the terms have to be better defined. I suggest that the author introduce the terms and associated equations to avoid ambiguity. Also, the rate constants can then be mentioned in figure 3a?

5) the sentence “The half-lives of light irradiation and thermal relaxation” is quite confusing. A

thermal half life is only depending on temperature, but an “irradiation half life” need to be defined more precisely, since it depends also on concentration, homogeneity of sample, light intensity, wavelength of irradiation etc.

6) the authors write ” perform light- induced anti-counterfeiting.” in essence, what they are presenting is an “anti-counterfeiting concept” this is a nice concept demonstration: but it is happening in solution. I suggest that they are more clear about this, maybe write something along the lines of: “” to demonstrate a possible future application of this dynamic photoswitch system, we developed an anti-counterfeiting proof of concept experiment in which...”

7) in the discussion, the authors use the term “to build up an intelligent multicolor luminescence system.” I think that they, in my view have to elaborate on the use of the term, or rewrite.

Minor comments:

The English writing could be improved for better flow, please find here selected comments related to this.

a) In the abstract: “simply photocontrol” should read “simple photocontrol”

b) in the intro reads ” especially the optical materials for colorful, ever-changing, and real- time display” should read “especially optical materials can be used to create colorful, ever-changing real-time displays”

c) “In another words” change to ”in other words”

d) what is meant with “photoluminescence-oriented and photoswitch-oriented groups ” I am not sure that I understand the meaning of these terms, please consider to reformulate.

e) “usually results undesired” should be ”usually results in undesired”

f) the sentence “Additionally, by means of photonic crystal systems, the composite or doped molecular photoswitches could drive luminescence shift more thoroughly, but such systems are hard to be confined in nanoscale or in well-defined nanostructures” need to be reformulated, it is difficult to read.

g) “To overcome such restrictions, we wonder if they could synergize another powerful tunable luminescence mechanism, excimer ” consider reformulation, the term “ we wonder” is not often used maybe use “we considered”

h) the statement “to form PSP and reaches unchanged for another ~110 s ” has some ambiguity, please re-formulate.

i) “PMC shows different photochromic fluorescence ranging of color” change to

Reviewer #2 (Remarks to the Author):

In this manuscript, Li, Chen and colleagues synthesized a variable molecular fluorescence emitter that exhibits a wide range of emission wavelengths, ranging from red to deep blue with a width of 240 nm. This was smartly achieved through the utilization of photoisomerization and excitation as two orthogonal wavelengths. The highlights of this study lie in the authors' molecular design approach, which involves linking pyrene and merocyanine in a conjugated manner to control the stacking status of pyrene. The authors conducted comprehensive experiments to investigate the ultrawide photoswitchable fluorescence behavior of their designed molecule, both in various solvents and films. Additionally, the authors demonstrated potential applications for PMC in anti-counterfeiting measures, photowriting techniques, and self-erasing systems. Overall, this work presents an interesting contribution with impressive experimental evidence. Therefore, I would like to recommend its publication in Nature Communications.

The authors should address the following minor issues:

-In Figure 1a (i) and (ii), it would be more helpful to mark the linkage between the molecular switches and fluorescent dyes for easier understanding. And the different states should be marked in the image for better understanding. Please check the Figure caption of Figure 1, “I n this work” should be “in this work”.

-In Figure 2, the color of "H4" and "H4'" differs from that of "H3" and "H3'", suggesting a potential error in the coloring of H4 and the marked line. The authors are advised to verify this discrepancy

-The reasonableness and comprehensibility of Figure 3a are evident; however, it would be advantageous to provide supporting references for both the figure and the corresponding statement.

-In Figure 7d, the authors reported a recovery of the fluorescence pattern after subjecting it to a dark period of 12 hours. However, comprehensive spectral data supporting this observation were not provided. It would be advantageous to include time-dependent fluorescence spectra as supplementary results to substantiate their claim.

-It is very interesting to see the photochromic fluorescence of PMC in different solvents, the authors are suggested to provide detailed explanations for this phenomenon.

- In the supplementary information, it is noteworthy that compound 1 appears to be previously unreported; however, the authors have only provided ¹H-NMR spectra. Therefore, inclusion of ¹³C-NMR and high-resolution mass spectra is recommended.

Reviewer #3 (Remarks to the Author):

The work describes the synthesis and spectroscopical and photochemical properties of an

essentially spiropyran-merocyanine photochromic system with the addition of a pyrene moiety. Interestingly, the reversibly photoconvertible molecule displays photochromism in absorption and in emission. As an application, the authors explore the use in anti counterfeiting function. The molecular system is well conceived and convincingly synthesized. Its emission photochromism is its main feature. Which linked to the practically complete conversion possibility to each form, provides a molecule with almost ideal signaling abilities.

There are two main mistakes in the experiments and in the interpretation of the results that must be amended before the work is considered for publication.

In photochemical experiments, the authors report as isomerization efficiency what is actually photoconversion extent. In photochemistry, the terms efficiency and yield are linked to the amount of absorbed photons. The authors do not specify features of the irradiation source: type or spectral distribution, only its power density. According to the high power density and irradiation times employed in the experiments, the photoconversion yield seems to be quite low.

There is no sense in reporting half lives in photoirradiations without knowing the absorbed photon flux. Furthermore there is no sense in comparing half lives in photochemical and thermal reactions. The authors should report quantum yields of photoisomerization.

Furthermore, the absorption of the system at the excitation wavelength in emission measurements is not reported. According the corresponding absorption spectra reported, it can be inferred that some emission spectra were performed at values of absorbance much greater than the upper threshold of 0.1, and maybe reaching absorbance values as high as 4. This can alter emission time profiles and spectral distribution (which turns SI Fig 8 doubtful).

The interpretation mistake is related to the assignment of the absorption band in the visible of the PMC form and the postulation of an excimer in this state.

According to IUPAC: Excimer: Complex formed by the interaction of an excited molecular entity with another identical molecular entity in its ground state. The complex dissociates in the ground state because it is “nonbonding” in the ground state.

Consequently, an excimer does not display an absorption band and its emission time profile exhibits a rising component with the concentration dependent lifetime of the monomer emission.

Whereas the authors assign an absorption band to the excimer, the time dependence of the excimer emission does not show a rising component (SI Fig.2). Moreover, the visible absorption band is typical of the merocyanine form and the authors report the expected typical negative solvatochromism of this type of compounds.

Other remarks are:

- No reference to former work in SP-MC photoisomerization mechanism and kinetic studies is made, as well as on quantum mechanical calculations.
- The term half life is misleading in the context of the work. The authors seem to be reporting the inverse of the first order decay rate constant, related to the half life by a $\ln 2$ factor.
- Fig 4c: MEH-Py does not correspond to any compound of the work.
- There are sentences containing various spelling and construction mistakes. Please correct.
- Reported numbers are incredibly precise. 5 – 10% uncertainty is typical in photochemical reaction efficiencies and quantum yields, even for standards.
- Fig 4b: Which is the excitation wavelength for the fluorescence?
- Figure 3 SI: Do the black spectra correspond to the photosteady state after irradiation at each wavelength. Seems doubtful because irradiation at wavelengths were only the PMC form absorbs

show increasing amounts of PSP (compare 550, 600, and 650 nm irradiation) Particularly at these two later wavelengths, the absorption of PMC is very low and the effect seems to be due to incomplete conversion. Therefore spectra are not comparable. For these reasons Table 1 should be eliminated. On ones side significant figures in the data are excessive. On the other side the reported efficiency has no sense. In photochemical reactions quantum yields should be reported.

- Figure 8 SI: Excitation wavelength for the emission spectra should be given. For what reason can the emission distribution change with concentration for PSP? Which is the absorption at the excitation wavelength of each solution. According to the presence of the narrow peak at around 740 nm, I guess excitation wavelength was probably 365 nm and the authors did not place a cut off filter to take the spectrum. If 365 nm was the excitation wavelength, then absorbance was too high to register an appropriate emission spectrum ($A(\lambda_{exc}) < 0.1$) if the authors did not use front face geometry, which is not stated in the methods.

Reviewer #4 (Remarks to the Author):

RE: Manuscript number: NCOMMS-23-53599

We are grateful to all the respected reviewers for their valuable time and helpful, encouraging comments. We have carefully revised our manuscript and Supplementary Information by taking into account the respected reviewers' comments as appropriate.

Our point-by-point response to the reviewers' comments and the changes made in our revised manuscript and Supplementary Information with red color highlighted track change are as follows.

Reviewer #1:

The manuscript "An ultrawide-range photochromic molecular fluorescence emitter" by Quan Li, Xu-Man Chen and co-workers considers photochromic fluorescent molecules with a broad spectral range.

*The essence of the paper is that the authors have created a photoswitch system that combine ICT state engineering with controlled excimer formation to create new advanced functional materials. In general **the work is interesting and well carried out**, my comments are mainly to the language and the use of specific terms such as "intelligent materials" which I personally think should be clarified."*

All together, I think that this is a very nice piece of work that deserves publication in Nature Commun, in my view, it has all the needed elements: it has a novel physical effect and very nice experimental verification. I however have quite a number of minor comments that I suggest that the authors look into, these comments are given with the best intention as feedback on possible ways on how to improve the manuscript further.

Our response: We sincerely thank the reviewer for his/her valuable time in going through our manuscript and providing the encouraging comments.

1) *The authors write "Here, we describe a synergistic intramolecular charge transfer (ICT) and excimer strategy in constructing photochromic molecular light emitter to enable three advancements in intelligent optical materials"*

I need to be convinced on the use of the term "intelligent" here I would prefer the term "functional"

Our response: Per suggestion, we have used "functional" instead of "intelligent" in our revised manuscript.

2) *figure 3 proposes a mechanism and calculates orbitals and energy levels of some of the involved species. To verify the proposed mechanism, I suggest to calculate the energies of all proposed intermediates and transition states and construct an energy-reaction coordinate plot. The terms T1-T3 are a bit difficult to understand here, are they transition states? Or transient species? By definition, you cannot have a conversion from a transition state to a transition state. Maybe what the authors mean is that T1-3 are intermediates? I suggest to remake figure 3 with correct TS and intermediate structures, and then revise the discussion accordingly. If the proposed mechanism is correct, there should be 3 intermediates, two photo isomers and 5 transition states. Or?*

Please clarify.

Our response: Per suggestion, we have added the calculation of energies of all proposed intermediates and transition states in Figure 3 accompanying with the description in our revised manuscript.

3) the authors use the term “The isomerization efficiency” is that the “photostationary state”? what is the quantum efficiency of conversion? Please clarify.

Our response: Per suggestion, we have replaced the term “The isomerization efficiency” into “conversion rate” in our revised manuscript. The quantum efficiency of conversion has also been determined and added into our revised Supplementary Information.

4) the sentence “The half-lives of light irradiation and thermal relaxation processes undergoes little changes in different cycles, and the average half-lives of the two processes are calculated (2.40 ± 0.24) s and (2000 ± 52) s” is very difficult to understand. In normal photoswitch systems, there can only be one thermal half life? The rates and the terms have to be better defined. I suggest that the author introduce the terms and associated equations to avoid ambiguity. Also, the rate constants can then be mentioned in figure 3a?

Our response: Per suggestion, we have added the definition of the term “half-life” in the Methods section and the rate constants in the caption of Figure 5a in our revised manuscript.

5) the sentence “The half-lives of light irradiation and thermal relaxation” is quite confusing. A thermal half life is only depending on temperature, but an “irradiation half life” need to be defined more precisely, since it depends also on concentration, homogeneity of sample, light intensity, wavelength of irradiation etc.

Our response: Per suggestion, we have given the more detailed description about these experiment in Methods into our revised manuscript.

6) the authors write “perform light- induced anti-counterfeiting.” in essence, what they are presenting is an “anti-counterfeiting concept” this is a a nice concept cemonstration: but it is happening in solution. I suggest that they are more clear about this, maybe write something along the lines of: “” to demonstrate a possible future application of this dynamic photoswitch system, we developed an anti-counterfeiting proof of concept experiment in which...”

Our response: Per suggestion, we have added the corresponding description into our revised manuscript.

7) in the discussion, the authors use the term “to build up an intelligent multicolor luminescence system.” I think that they, in my view have to elaborate on the use of the term, or rewrite.

Our response: Per suggestion, we have rewritten this sentence in our revised manuscript.

Minor comments:

The English writing could be improved for better flow, please find here selected comments related to this.

Our response: Per suggestion, we have carefully polished the English writing in our revised manuscript.

a) In the abstract: “simply photocontrol” should read “simple photocontrol”

Our response: We have corrected the typo in our revised manuscript.

b) in the intro reads ” especially the optical materials for colorful, ever-changing, and real- time display” should read “especially optical materials can be used to create colorful, ever-changing real-time displays”

Our response: Per suggestion, we have corrected it in our revised manuscript.

c) “In another words” change to “in other words”

Our response: We have corrected the typo in our revised manuscript.

d) what is meant with “photoluminescence-oriented and photoswitch-oriented groups ” I am not sure that I understand the meaning of these terms, please consider to reformulate.

Our response: Per suggestion, we have rewritten it in our revised manuscript.

e) “usually results undesired” should be “usually results in undesired”

Our response: Per suggestion, we have corrected it in our revised manuscript.

f) the sentence “Additionally, by means of photonic crystal systems, the composite or doped molecular photoswitches could drive luminescence shift more thoroughly, but such systems are hard to be confined in nanoscale or in well-defined nanostructures” need to be reformulated, it is difficult to read.

Our response: Per suggestion, we have rewritten it in our revised manuscript.

g) “To overcome such restrictions, we wonder if they could synergize another powerful tunable luminescence mechanism, excimer ” consider reformulation, the term “ we wonder” is not often used maybe use “we considered”

Our response: Per suggestion, we have corrected it in our revised manuscript.

h) the statement “to form PSP and reaches unchanged for another ~110 s ” has some ambiguity, please re-formulate.

Our response: Per suggestion, we have re-formulated it in our revised manuscript.

i) “PMC shows different photochromic fluorescence ranging of color ” change to

Our response: We have rewritten it in our revised manuscript.

Reviewer #2:

In this manuscript, Li, Chen and colleagues synthesized a variable molecular fluorescence emitter that exhibits a wide range of emission wavelengths, ranging from red to deep blue with a width of 240 nm. This was smartly achieved through the utilization of photoisomerization and excitation as two orthogonal wavelengths. The highlights of this study lie in the authors' molecular design approach, which involves linking pyrene and merocyanine in a conjugated manner to control the stacking status of pyrene. The authors conducted comprehensive experiments to investigate the ultrawide photoswitchable fluorescence behavior of their designed molecule, both in various solvents and films. Additionally, the authors demonstrated potential applications for PMC in anti-counterfeiting measures, photowriting techniques, and self-erasing systems. Overall, this work presents an interesting contribution with impressive experimental evidence. Therefore, I would like to recommend its publication in Nature Communications.

Our response: We sincerely thank the reviewer for his/her valuable time in going through our manuscript and providing the encouraging comments.

The authors should address the following minor issues:

-In Figure 1a (i) and (ii), it would be more helpful to mark the linkage between the molecular switches and fluorescent dyes for easier understanding. And the different states should be marked in the image for better understanding. Please check the Figure caption of Figure 1, "In this work" should be "in this work".

Our response: Per suggestion, we have added the marks into Figure 1a and corrected the typo in our revised manuscript.

-In Figure 2, the color of "H4" and "H4'" differs from that of "H3" and "H3'", suggesting a potential error in the coloring of H4 and the marked line. The authors are advised to verify this discrepancy

Our response: Per suggestion, we have corrected the typo of Figure 2 in our revised manuscript.

-The reasonableness and comprehensibility of Figure 3a are evident; however, it would be advantageous to provide supporting references for both the figure and the corresponding statement.

Our response: Per suggestion, we have updated the Figure 3 and rewritten the corresponding statement in our revised manuscript.

-In Figure 7d, the authors reported a recovery of the fluorescence pattern after subjecting it to a dark period of 12 hours. However, comprehensive spectral data supporting this observation were not provided. It would be advantageous to include time-dependent fluorescence spectra as supplementary results to substantiate their claim.

Our response: We have added the suggested spectra in our revised Supplementary Information.

-It is very interesting to see the photochromic fluorescence of PMC in different solvents, the authors are suggested to provide detailed explanations for this phenomenon.

Our response: Per suggestion, we have added the detailed explanation for this phenomenon into our revised manuscript.

- In the supplementary information, it is noteworthy that compound 1 appears to be previously unreported; however, the authors have only provided ¹H-NMR spectra. Therefore, inclusion of ¹³C-NMR and high-resolution mass spectra is recommended.

Our response: We have added ¹³C-NMR and high-resolution mass spectra in our revised Supplementary Information.

Reviewer #3:

The work describes the synthesis and spectroscopical and photochemical properties of an essentially spiropyran-merocyanine photochromic system with the addition of a pyrene moiety. Interestingly, the reversibly photoconvertible molecule displays photochromism in absorption and in emission. As an application, the authors explore the use in anti counterfeiting function.

The molecular system is well conceived and convincingly synthesized. Its emission photochromism is its main feature. Which linked to the practically complete conversion possibility to each form, provides a molecule with almost ideal signaling abilities.

Our response: We sincerely thank the reviewer for his/her valuable time in going through our manuscript and providing the encouraging comments.

There are two main mistakes in the experiments and in the interpretation of the results that must be amended before the work is considered for publication.

In photochemical experiments, the authors report as isomerization efficiency what is actually photoconversion extent. In photochemistry, the terms efficiency and yield are linked to the amount of absorbed photons. The authors do not specify features of the irradiation source: type or spectral distribution, only its power density. According to the high power density and irradiation times employed in the experiments, the photoconversion yield seems to be quite low.

There is no sense in reporting half lives in photoirradiations without knowing the absorbed photon flux. Furthermore there is no sense in comparing half lives in photochemical and thermal reactions. The authors should report quantum yields of photoisomerization.

Our response: Per suggestion, we have replaced the term “The isomerization efficiency” into “conversion rate” in our revised manuscript. The quantum yields of photoisomerization have been added into our revised Supplementary Information.

Furthermore, the absorption of the system at the excitation wavelength in emission measurements is not reported. According the corresponding absorption spectra reported, it can be inferred that some emission spectra were performed at values of absorbance much greater than the upper threshold of 0.1, and maybe reaching

absorbance values as high as 4. This can alter emission time profiles and spectral distribution (which turns SI Fig 8 doubtful).

Our response: The excitation wavelength has added into the caption of SI Fig. 8. Actually, we used a series of quartz cell with different light path to ensure that the absorbance of **PSP** at excitation wavelength is about 0.1 as shown in our SI Fig. 9. Thus, we obtained the UV-Vis absorption and fluorescence emission spectra of **PSP** in different concentration as shown as our SI Fig. 8. The emission variation with the concentration of **PSP** may result from the presence of a small amount of **PSP** excimer at its excited state. With a higher concentration of **PSP**, a large amount of the monomer results in the emission blue shifting.

The interpretation mistake is related to the assignment of the absorption band in the visible of the PMC form and the postulation of an excimer in this state.

According to IUPAC: Excimer: Complex formed by the interaction of an excited molecular entity with another identical molecular entity in its ground state. The complex dissociates in the ground state because it is “nonbonding” in the ground state. Consequently, an excimer does not display an absorption band and its emission time profile exhibits a rising component with the concentration dependent lifetime of the monomer emission. Whereas the authors assign an absorption band to the excimer, the time dependence of the excimer emission does not show a rising component (SI Fig.2). Moreover, the visible absorption band is typical of the merocyanine form and the authors report the expected typical negative solvatochromism of this type of compounds.

Our response: Per suggestion, we have comprehensively measured the lifetimes of **PMC** at the two emission states by controlling the concentration in CHCl_3 (Ex: 365 nm, Em: 640 nm for 10^{-4} M and 400 nm for 10^{-5} M) which have been added in our revised Supplementary Fig. 4b. The lifetimes of these **PMC** solutions are calculated 1.71 ns and 1.39 ns at 640 nm and 400 nm, respectively, indicating that the lifetime of 640 nm emission (10^{-4} M) is significantly longer than that of 400 nm emission (10^{-5} M). These results demonstrate that the 640 nm emission of **PMC** at a higher concentration tends to be an excimer emission while the 400 nm emission of **PMC** is the monomer emission state, exhibiting a rising component with the concentration dependent lifetime of the monomer emission. On the other hand, we recognize that the formation of excimer depends on the interaction of an excited molecular entity with another ground-state molecular entity, and the excimer dissociates in the ground state. As shown in the updated Supplementary Fig. 4, the concentration-dependent UV-Vis spectra of **PMC** ranging from 0.0001 mM to 0.1 mM in CHCl_3 shows no difference in absorption band, indicating **PMC** is “nonbonding” in the ground state. The solvatochromism of **PMC** showing in UV-Vis spectra belongs to its dissociated and aggregated state in different solvents. The aggregated state of **PMC** in CHCl_3 demonstrate that **PMC** have potential to form excimer, but it does not represent any specific dimer-like interaction. The corresponding description has also been added into our revised manuscript and Supplementary Information.

Other remarks are:

- No reference to former work in SP-MC photoisomerization mechanism and kinetic studies is made, as well as on quantum mechanical calculations.

Our response: Per suggestion, we have added quantum mechanical calculations in Figure 3 and its corresponding reference in our revised manuscript.

-The term half life is misleading in the context of the work. The authors seem to be reporting the inverse of the first order decay rate constant, related to the half life by a $\ln 2$ factor.

Our response: Per suggestion, we have defined the term “half-life” in the Methods section in our revised manuscript.

- Fig 4c: MEH-Py does not correspond to any compound of the work.

Our response: Per suggestion, we have corrected it in our revised manuscript.

-There are sentences containing various spelling and construction mistakes. Please correct.

Our response: Per suggestion, we have carefully checked and revised the spelling and construction mistakes in our revised manuscript.

- Reported numbers are incredibly precise. 5 – 10% uncertainty is typical in photochemical reaction efficiencies and quantum yields, even for standards.

Our response: Per suggestion, we have corrected the precision of photochemical reaction efficiencies and quantum yields in our revised manuscript.

- Fig 4b: Which is the excitation wavelength for the fluorescence?

Our response: The excitation wavelength is 365 nm for the fluorescence in this work unless mentioned.

- Figure 3 SI: Do the black spectra correspond to the photosteady state after irradiation at each wavelength. Seems doubtful because irradiation at wavelengths where only the PMC form absorbs show increasing amounts of PSP (compare 550, 600, and 650 nm irradiation) Particularly at these two later wavelengths, the absorption of PMC is very low and the effect seems to be due to incomplete conversion. Therefore spectra are not comparable. For these reasons Table 1 should be eliminated. On one side significant figures in the data are excessive. On the other side the reported efficiency has no sense. In photochemical reactions quantum yields should be reported.

Our response: We confirm that the black spectra correspond to the photosteady state at our experimental condition that PMC and PSP reached a photo-equilibrium. Additionally, we have added the experiments of photochemical reactions quantum yields in our revised Supplementary Information as the updated Supplementary Fig.3. and Table 1.

- Figure 8 SI: Excitation wavelength for the emission spectra should be given. For what reason can the emission distribution change with concentration for PSP? Which is the

absorption at the excitation wavelength of each solution. According to the presence of the narrow peak at around 740 nm, I guess excitation wavelength was probably 365 nm and the authors did not place a cut off filter to take the spectrum. If 365 nm was the excitation wavelength, then absorbance was too high to register an appropriate emission spectrum ($A(\lambda_{exc}) < 0.1$) if the authors did not use front face geometry, which is not stated in the methods.

Our response: The excitation wavelength is 365 nm which has been added into the Methods section in our revised manuscript. Also, as we mentioned above, we used a series of quartz cell with different light path to ensure that the absorbance of **PSP** at excitation wavelength is about 0.1 (SI Fig. S9). Thus, we obtained the UV-Vis absorption and fluorescence emission spectra of **PSP** in different concentration (SI Fig. S8). The fluorescence emission spectra exhibits that the wavelengths of the emission maxima are all around 400 nm.

Reviewer #4:

Our response: We sincerely thank the reviewer for his/her valuable time in going through our manuscript and providing the encouraging comments.

Furthermore, we have carefully checked the manuscript and the Supplementary Information. With these changes and point-by-point response to the reviewers' comments, we hope that the revised manuscript is now acceptable for publication.

Your kind consideration of the revised manuscript will be greatly appreciated.

REVIEWER COMMENTS

Reviewer #1 (Remarks to the Author):

The authors have done a good job in revising the manuscript, and I can now recommend publication.

The manuscript still need some language editing, but I assume that this can be done during the editorial process. One example:

"We fitted the variation off absorbance at 500 nm of five light irradiation and thermal relaxation processes to find that two process belonging to first-order reaction". Should read "We fitted the variation off absorbance at 500 nm of five light irradiation and thermal relaxation processes to find that the two processes operate according to first-order reaction kinetics."

Reviewer #2 (Remarks to the Author):

This work demonstrates the unique nature of an ultrawide-range photochromic molecular fluorescence emitter and its anti-counterfeiting function. The authors responded to most of the reviewers' comments, and the manuscript was properly revised. I think this manuscript can be published in Nature Communications.

Reviewer #3 (Remarks to the Author):

See attached file

[Editorial Note: This file is displayed across the next three pages]

This revised version does not show either the expected improvement in the paper or convincing reply to the raised doubts, as detailed in what follows.

The lifetime of PMC at 640 nm (0.1 mM) is measured to be 1.71 ns, which is much longer than that at 400 nm (0.01 mM, 1.39 ns), confirming that the 640 nm and 400 nm emission of PMC belongs to excimer and monomer state, respectively (Supplementary Fig. 4b).

This is not a proof of the excimer. No build up time is observed or reported. The difference in emission lifetime is not a convincing argument. The emission at 0.01 mM is not a single exponential. The longer time component of this decay (Fig. S4b), though having an amplitude 10 times smaller than the short time component, has a lifetime (not informed) ca. 10 times longer, thus contributing equally to the total steady state emission. Excitation and emission wavelengths should be added to the caption of Figure S4b.

Per suggestion, we have replaced the term “The isomerization efficiency” into “conversion rate” in our revised manuscript.

In this case conversion rate is not correct. The authors inform a dimensionless value. A rate should have units of inverse time. Photoisomerization extent is the correct name for the value informed. This should be corrected throughout.

The excitation wavelength has added into the caption of SI Fig. 8. Actually, we used a series of quartz cell with different light path to ensure that the absorbance of PSP at excitation wavelength is about 0.1 as shown in our SI Fig. 9. Thus, we obtained the UV-Vis absorption and fluorescence emission spectra of PSP in different concentration as shown as our SI Fig. 8.

This is an important experimental condition which should be added to the Fluorescence Spectroscopy paragraph of the Methods section.

The emission variation with the concentration of PSP may result from the presence of a small amount of PSP excimer at its excited state. With a higher concentration of PSP, a large amount of the monomer results in the emission blue shifting.

This reply is confusing. The amount of excimer increases with concentration and the emission shoulder at ca. 460 nm is observed only at low concentrations. Furthermore, the emission at 640 nm is assigned to the PMC excimer by the authors and there is no evidence of this emission in Figure S8. In this answer, the authors refer to a PSP excimer, which was not discussed nor introduced in any passage of the manuscript. The emission at 460 nm has other origin.

Per suggestion, we have comprehensively measured the lifetimes of PMC at the two emission states by controlling the concentration in CHCl₃ (Ex: 365 nm, Em: 640 nm for 10⁻⁴ M and 400 nm for 10⁻⁵ M) which have been added in our revised Supplementary Fig. 4b. The lifetimes of these PMC solutions are calculated 1.71 ns and 1.39 ns at 640 nm and 400 nm, respectively, indicating that the lifetime of 640 nm emission (10⁻⁴ M) is significantly longer than that of 400 nm emission (10⁻⁵ M). These results demonstrate that the 640 nm emission of PMC at a higher concentration tends to be an excimer emission while the 400 nm emission of PMC is the monomer emission state, exhibiting a rising component with the concentration dependent lifetime of the monomer emission.

The decays in Figure S2 or S4b do not show any build up component. Do the authors have other data to show that there is a build up with the lifetime of the monomer? In such case, they should display them as well as the sum of exponentials fit and this evidence should be in the main text. This is a crucial evidence for an excimer.

As concentration increases, the authors mention in their reply that they correctly used cuvettes of shorter light path. Now cuvettes of 0.1 or 0.01 cm as they were forced to use, necessarily imply the measurement in front face configuration, which is difficult to quantify and very vulnerable to self absorption effects, influencing more the 400 nm band, where the compound strongly absorbs, than the 640 nm band, where the compound does not absorb.

On the other hand, we recognize that the formation of excimer depends on the interaction of an excited molecular entity with another ground-state molecular entity, and the excimer dissociates in the ground state. As shown in the updated Supplementary Fig. 4, the concentration-dependent UV-Vis spectra of PMC ranging from 0.0001 mM to 0.1 mM in CHCl₃ shows no difference in absorption band, indicating PMC is “nonbonding” in the ground state.

The spectra of Figure S4a and its corresponding linear absorbance dependence with concentration displayed in Figure 4a represent the expected behavior for a single species: the open PMC form of the compound. This is in favor of the assignment of this band to the PMC monomer. This is the simplest interpretation. The author’s interpretation is too speculative.

The solvatochromism of PMC showing in UV-Vis spectra belongs to its dissociated and aggregated state in different solvents. The aggregated state of PMC in CHCl₃ demonstrate that PMC have potential to form excimer, but it does not represent any specific dimer-like interaction. The corresponding description has also been added into our revised manuscript and Supplementary Information.

But Figure 4a demonstrates that PMC is not aggregated in chloroform.

The second part of Table S1 should be eliminated. The complete kinetic characterization of the system is provided by the quantum yield of photoisomerization. The authors calculate this value from the initial photoisomerization rate and the absorbed photon flux. The description of the photoisomerization kinetics by a single exponential is only an approximation when the fraction of light absorbed is small and Lambert Beers law can be linearized. The first order rate constant derived from such an analysis should be:

$$k = \phi \cdot I_0 \cdot \ln 10 \cdot \epsilon \cdot l = \phi \cdot I_a / C$$

K(reported) (1/s)	phi	I _a (uM/s)	C (uM)	phi*I _a /C (1/s)
0,12	0,3	38	96	0,12
0,33	0,64	50	98	0,33
0,22	0,4	53	96	0,22
0,27	0,31	82,4	96	0,27
0,038	0,041	89,7	98	0,038
0,0059	0,006	98,2	96	0,006

The first column, values obtained from the fits to the decays of Fig. S3 is identical to the last column, derived from data of the first part according to the expression of the first order rate constant.).

Figure S23: In the expressions of the fitting equations, the time units of each fit should be given. Seem to be 1/s but for the photoisomerization irradiation wavelength and initial absorbed photon flux should be given.

Other points that were raised and do not have a satisfactory reply are:

i) The persistence of informing half lives for photochemical reactions. This should be corrected because it is a meaningless value, as explained above.

Photoisomerization efficiency persists in Figure 5c, Figure S11f, and Figure S12f.

ii) Conversion rate in % should be changed throughout as suggested. What is informed as conversion rate is conversion extent, dependent on absorbed photon flux and irradiation time.

iii) Figure S11. It provides no new information. Solvent, irradiation wavelength and incident photon flux should be informed.

iv) Insufficient details on experimental conditions, such as irradiation wavelength and photon flux, cuvettes, light path, and configuration (whether normal 90° or front face arrangement) used for emission experiments.

In conclusion, the compound is interesting by its dual photochromism in absorption and in emission. The authors insist in their interpretation of the emission by an excimer of which there is no convincing evidence. On the contrary, the absorption in the green and the emission in the red, as well as their negative solvatochromism, is a common feature of the merocyanine form of these type of compounds. This fact, the many mistakes in magnitude description, such as half lives for photochemical reactions without further explanation, conversion rates for conversion extent, photoisomerization efficiency for conversion extent, as well as uncorrected language mistakes (e.g. *“The quantum efficiency of these light is also calculated”*, *“Quantum yield and fluorescence lifetime of PMC and PSP...”* without explicitly referring to “Emission quantum yield”, among many other mistakes) that turn understanding more difficult, greatly demerit the work. This revised version does not show a great improvement compared to the original submission. In the present form, publication is not recommended.

Reviewer #4 (Remarks to the Author):

I co-reviewed the revised version of this manuscript with one of the reviewers who provided the listed reports.

RE: Manuscript number: NCOMMS-23-53599

We are grateful to all the respected reviewers for their helpful comments. We sincerely thank two of the reviewers to their recommendation of its publication. Moreover, we have carefully revised our manuscript and Supplementary Information by taking into account the respected reviewers' comments as appropriate.

Our point-by-point response to the four reviewers' comments and the changes made in our revised manuscript and Supplementary Information are as follows. Also, we are attaching the track changed manuscript with its SI for your kind review.

Reviewer #1

The authors have done a good job in revising the manuscript, and I can now recommend publication.

Our response: We sincerely thank the reviewer for his/her valuable time in going through our manuscript and providing the encouraging comments.

The manuscript still need some language editing, but I assume that this can be done during the editorial process. One example: "We fitted the variation off absorbance at 500 nm of five light irradiation and thermal relaxation processes to find that two process belonging to first-order reaction". Should read "We fitted the variation off absorbance at 500 nm of five light irradiation and thermal relaxation processes to find that the two processes operate according to first-order reaction kinetics."

Our response: Per suggestion, we have carefully checked and revised the language mistakes in our revised manuscript.

Reviewer #2

This work demonstrates the unique nature of an ultrawide-range photochromic molecular fluorescence emitter and its anti-counterfeiting function. The authors responded to most of the reviewers' comments, and the manuscript was properly revised. I think this manuscript can be published in Nature Communications.

Our response: We sincerely thank the reviewer for his/her valuable time in going through our manuscript and providing the encouraging comments.

Reviewer #3

This revised version does not show either the expected improvement in the paper or convincing reply to the raised doubts, as detailed in what follows.

Our response: We sincerely thank the reviewer for his/her valuable time in going through our manuscript and providing the helpful comments. We feel sorry that the previously revised version was not satisfactory for the reviewer. We hope this revised version with the following new results and improvement can be the convincing reply to

the questions or concerns from the respected reviewer.

The lifetime of PMC at 640 nm (0.1 mM) is measured to be 1.71 ns, which is much longer than that at 400 nm (0.01 mM, 1.39 ns), confirming that the 640 nm and 400 nm emission of PMC belongs to excimer and monomer state, respectively (Supplementary Fig. 4b).

This is not a proof of the excimer. No build up time is observed or reported. The difference in emission lifetime is not a convincing argument. The emission at 0.01 mM is not a single exponential. The longer time component of this decay (Fig. S4b), though having an amplitude 10 times smaller than the short time component, has a lifetime (not informed) ca. 10 times longer, thus contributing equally to the total steady state emission. Excitation and emission wavelengths should be added to the caption of Figure S4b.

Our response: According to the references *J. Phys. Chem. Lett.* 2018, **9**, 2138 and *J. Phys. Chem. A* 2020, **124**, 8478, we have investigated the life decay and time-resolved fluorescence (TRF) spectra for the PMC in CHCl₃ (0.03 mM), shown in our revised Supplementary Information as update Fig. S4. The emission at 640 nm of PMC has a noticeable rise time of ca. 2.2 ns, while the rise time of emission at 400 nm is ca. 1.6 ns. Therefore, we think the longer rise time (~1.4 times) for the emission at 640 nm of PMC may result from the build-up time of excimer, which in accordance with the TRF spectra for the PMC in CHCl₃.

Per suggestion, we have replaced the term “The isomerization efficiency” into “conversion rate” in our revised manuscript.

In this case conversion rate is not correct. The authors inform a dimensionless value. A rate should have units of inverse time. Photoisomerization extent is the correct name for the value informed. This should be corrected throughout.

Our response: Per suggestion, we have corrected the related term in our revised manuscript.

The excitation wavelength has added into the caption of SI Fig. 8. Actually, we used a series of quartz cell with different light path to ensure that the absorbance of PSP at excitation wavelength is about 0.1 as shown in our SI Fig. 9. Thus, we obtained the UV-Vis absorption and fluorescence emission spectra of PSP in different concentration as shown as our SI Fig. 8.

This is an important experimental condition which should be added to the Fluorescence Spectroscopy paragraph of the Methods section.

Our response: The UV-Vis and emission results for the PSP in CHCl₃ by a series of quartz cells with different light paths are shown here. The experimental condition that we selected is the quartz cells with light path like 5 mm, 2 mm, 1 mm and 0.5 mm to control the absorbance at 365 nm of PSP in CHCl₃ near 0.1 Abs. The emission spectra are obtained with a suitable quartz cell for the PSP in CHCl₃ with certain concentrations via normal 90° method. For example, the light path for the 0.1 mM PSP solution is 0.5 mm.

Fig. R1 **a**, UV-Vis absorption spectra of **PSP** in different concentrations. The absorbance at 365 nm of all **PSP** samples is controlled about 0.1 by adjusting the light path. **b**, The fluorescence emission spectra of the corresponding **PSP** samples of **a**.

The emission variation with the concentration of PSP may result from the presence of a small amount of PSP excimer at its excited state. With a higher concentration of PSP, a large amount of the monomer results in the emission blue shifting.

This reply is confusing. The amount of excimer increases with concentration and the emission shoulder at ca. 460 nm is observed only at low concentrations. Furthermore, the emission at 640 nm is assigned to the PMC excimer by the authors and there is no evidence of this emission in Figure S8. In this answer, the authors refer to a PSP excimer, which was not discussed nor introduced in any passage of the manuscript. The emission at 460 nm has other origin.

Our response: The emission spectra of PSP with varied concentrations in Fig. S11 were normalized. The shoulder peak at 460 nm can be found in emission spectra for all emission spectra. Therefore, shoulder peak at 460 nm may result from the presence of a small amount of PSP excimer at its excited state. With the concentration of PSP monomer increasing, the emission peak of monomer increased quickly to mask this shoulder peak from PSP excimer.

Per suggestion, we have comprehensively measured the lifetimes of PMC at the two emission states by controlling the concentration in CHCl₃ (Ex: 365 nm, Em: 640 nm for 10⁻⁴ M and 400 nm for 10⁻⁵ M) which have been added in our revised Supplementary Fig. 4b. The lifetimes of these PMC solutions are calculated 1.71 ns and 1.39 ns at 640 nm and 400 nm, respectively, indicating that the lifetime of 640 nm emission (10⁻⁴ M) is significantly longer than that of 400 nm emission (10⁻⁵ M). These results demonstrate that the 640 nm emission of PMC at a higher concentration tends to be an excimer emission while the 400 nm emission of PMC is the monomer emission state, exhibiting a rising component with the concentration dependent lifetime of the monomer emission.

The decays in Figure S2 or S4b do not show any build up component. Do the authors have other data to show that there is a build up with the lifetime of the monomer? In such case, they should display them as well as the sum of exponentials fit and this evidence should be in the main text. This is a crucial evidence for an excimer.

Our response: We added the related description in our revised manuscript and Figure S4 in our revised Supplementary Information. The main purpose and highlight of this article are that we used a single molecular switch to design an emitter with time-dependent multi-emission color changing via external stimuli in solution or film. We are not skilled in investigating the mechanism of emission because our major is material science and the equipment in our lab does not support us to investigate the mechanism more clearly in a short time. Therefore, we think that the Supplementary Information is suitable for placing these figures.

As concentration increases, the authors mention in their reply that they correctly used cuvettes of shorter light path. Now cuvettes of 0.1 or 0.01 cm as they were forced to use, necessarily imply the measurement in front face configuration, which is difficult to quantify and very vulnerable to self absorption effects, influencing more the 400 nm band, where the compound strongly absorbs, than the 640 nm band, where the compound does not absorb.

Our response: The cuvette that used for all absorption and emission experiments for PMC in solution is 1 cm and all emission spectra in our manuscript are obtained via normal 90° method.

On the other hand, we recognize that the formation of excimer depends on the interaction of an excited molecular entity with another ground-state molecular entity, and the excimer dissociates in the ground state. As shown in the updated Supplementary Fig. 4, the concentration-dependent UV-Vis spectra of PMC ranging from 0.0001 mM to 0.1 mM in CHCl₃ shows no difference in absorption band, indicating PMC is “nonbonding” in the ground state.

The spectra of Figure S4a and its corresponding linear absorbance dependence with concentration displayed in Figure 4a represent the expected behavior for a single species: the open PMC form of the compound. This is in favor of the assignment of this band to the PMC monomer. This is the simplest interpretation. The author’s interpretation is too speculative.

Our response: The signal broadening in NMR spectra is evidence of aggregation (*Chem. Eur. J.* 2012, **18**, 13665–13677). The NMR spectra of PMC in CDCl₃ is broadened when compared to the NMR spectra in DMSO-d₆. Considering this adverse factor, we select the CD₃OD as the solvent for obtaining the NMR spectra of PMC with different concentrations. The peak signal of PMC in low field shows the signal attenuation with the higher concentration to the lower concentration as well as some peaks shifting. Due to the sensitivity of each proton to its magnetic environment, observation of resonance shifts provides structural information about the self-assembled aggregates. (*Chem. Eur. J.* 2012, **18**, 13665–13677). It means that PMC is easily aggregated in solvent with a particular aggregate concentration.

Fig. R2 $^1\text{H-NMR}$ spectra of PMC in CD_3OD at different concentrations.

The solvatochromism of PMC showing in UV-Vis spectra belongs to its dissociated and aggregated state in different solvents. The aggregated state of PMC in CHCl_3 demonstrate that PMC have potential to form excimer, but it does not represent any specific dimer-like interaction. The corresponding description has also been added into our revised manuscript and Supplementary Information.

But Figure 4a demonstrates that PMC is not aggregated in chloroform.

Our response: In Fig.4c, two-stage variation of transmittance at 625 nm of PMC in CHCl_3 has already demonstrated the formation of the self-assembly, which does not necessarily affect the UV absorption in low concentration. The SEM image of PMC also demonstrates the formation of the self-assembly or aggregates.

The second part of Table S1 should be eliminated. The complete kinetic characterization of the system is provided by the quantum yield of photoisomerization. The authors calculate this value from the initial photoisomerization rate and the absorbed photon flux. The description of the photoisomerization kinetics by a single exponential is only an approximation when the fraction of light absorbed is small and Lambert Beers law can be linearized. The first order rate constant derived from such an analysis should be: $k = \phi \cdot I_0 \cdot I_n \cdot 10^{-\epsilon \cdot l} = \phi \cdot I_a / C$

K(reported) (1/s)	phi	Ia (uM/s)	C (uM)	phi*Ia/C (1/s)
0,12	0,3	38	96	0,12
0,33	0,64	50	98	0,33
0,22	0,4	53	96	0,22
0,27	0,31	82,4	96	0,27
0,038	0,041	89,7	98	0,038
0,0059	0,006	98,2	96	0,006

The first column, values obtained from the fits to the decays of Fig. S3 is identical to the last column, derived from data of the first part according to the expression of the first order rate constant.)

Our response: Per suggestion, we have update the Table S1 in our revised Supplementary Information.

Figure S23: In the expressions of the fitting equations, the time units of each fit should be given. Seem to be 1/s but for the photoisomerization irradiation wavelength and initial absorbed photon flux should be given.

Our response: Per suggestion, we have added the related description and contents in our revised Supplementary Information.

Other points that were raised and do not have a satisfactory reply are: i) The persistence of informing half lives for photochemical reactions. This should be corrected because it is a meaningless value, as explained above.

Our response: Per suggestion, half lives have not important significance to evaluate the photochemical reactions, so we move the related data into Supplementary Information.

Photoisomerization efficiency persists in Figure 5c, Figure S11f, and Figure S12f.

Our response: Per suggestion, we have updated these figures.

ii) Conversion rate in % should be changed throughout as suggested. What is informed as conversion rate is conversion extent, dependent on absorbed photon flux and irradiation time.

Our response: Per suggestion, we have corrected the related term as “photoisomerization extent” in our revised manuscript and Supplementary information.

iii) *Figure S11. It provides no new information. Solvent, irradiation wavelength and incident photon flux should be informed.*

Our response: Per suggestion, we have added the related description in our revised Supplementary Information Figure S11.

iv) *Insufficient details on experimental conditions, such as irradiation wavelength and photon flux, cuvettes, light path, and configuration (whether normal 90° or front face arrangement) used for emission experiments.*

Our response: Per suggestion, we have added the related description in the Method section.

In conclusion, the compound is interesting by its dual photochromism in absorption and in emission.

Our response: We sincerely thank that the reviewer endorses our innovation.

The authors insist in their interpretation of the emission by an excimer of which there is no convincing evidence.

Our response: We have updated the evidence for the excimer.

On the contrary, the absorption in the green and the emission in the red, as well as their negative solvatochromism, is a common feature of the merocyanine form of these type of compounds.

Our response: Importantly, our innovation in this work is using a single molecular photoswitch to design an emitter with photochromism in emission for achieving a broad range emission wavelength changing via external stimuli. Based on the photochromism of this emitter, we can effectively use this property to display security patterns in solution and film.

This fact, the many mistakes in magnitude description, such as half lives for photochemical reactions without further explanation, conversion rates for conversion extent, photoisomerization efficiency for conversion extent, as well as uncorrected language mistakes (e.g. “The quantum efficiency of these light is also calculated”, “Quantum yield and fluorescence lifetime of PMC and PSP...” without explicitly referring to “Emission quantum yield”, among many other mistakes) that turn understanding more difficult, greatly demerit the work.

Our response: Per suggestion, we have corrected the description in our revised manuscript.

This revised version does not show a great improvement compared to the original submission. In the present form, publication is not recommended.

Our response: We update the manuscript and supplementary information according to the reviewer's advice, and we hope this revised version can be acceptable for publication in *Nature Communications*.

Reviewer #4

I co-reviewed the revised version of this manuscript with one of the reviewers who provided the listed reports.

Our response: We sincerely thank the reviewer for his/her valuable time in going through our manuscript and providing the encouraging comments.

Furthermore, we have carefully checked the manuscript and its Supplementary Information. With these changes and point-by-point response to the reviewers' comments, we hope that the revised manuscript is now acceptable for publication.

Your kind consideration of the revised manuscript will be greatly appreciated.

REVIEWERS' COMMENTS

Reviewer #1 (Remarks to the Author):

[Note from the Editor: Reviewer #1 was asked to look over the response given to reviewer #3.]

My comments have been addressed, in my view, the manuscript can be accepted.

after correspondence with the editor, I have taken a second look:

I have read the author response again related to the “eximer formation” points raised by reviewer 3. Please note that I am not an expert in photophysics, but:

The answers by the authors seems solid enough.

Their argument that NMR is indication of aggregates is correct, but not necessarily eximer.

They have added extra details in the spectroscopy, and the discussion about eximer is not critical for the main points of the manuscript, and are in general uncontroversial e.g. it is normal that this class of molecules can make eximers.

All together, I think that it would be ok to accept the manuscript in the current form.

Reviewer #2 (Remarks to the Author):

[Note from the Editor: Reviewer #2 was asked to look over the response given to reviewer #3.]

This paper has undergone two rounds of review, and the comments from Reviewer 3 were not really that far apart in terms of mechanism investigation.

My own reading of this paper, is that the design of the compound in this work is interesting by its dual photochromism in absorption and in emission. The authors further explore its ultrawide photoswitchable fluorescence behavior and potential applications. I also agree with the comments and suggestions from Reviewer 3 that addressing the mechanism of emission would enhance the quality of this paper. Additionally, I understand the authors' statement about their limitations in investigating the mechanism more thoroughly within a short timeframe due to their focus on material science and limited laboratory equipment.

Given the complexity of the system, where spectral signals overlap and obscure each other, it is challenging for me to provide a definitive answer to the editor's query regarding the evidence supporting the formation of an excimer in the process. I hope that the editor can comprehend this difficulty and exercise their judgment on this issue. Therefore, I would like to defer to the editor's decision.

RE: Manuscript number: NCOMMS-23-53599B

We are grateful to all the editorial colleagues as well as all the respected reviewers for their valuable time and helpful comments. We agree with your viewpoint, therefore, after careful consideration, we have toned down the claims of “excimer formation” in our revised manuscript. The novelty and main design in our manuscript is the tunable dual photochromism in absorption and in emission rather than the formation of excimer, which is pointed out by both two reviewers. We have checked out and modified carefully our manuscript and Supplementary information as appropriate and provided all files according to the Author Checklist.

Our point-by-point response to the two reviewers’ comments and the editorial advice, and the changes made in our revised manuscript and Supplementary Information are as follows.

Reviewer #1

“My comments have been addressed, in my view, the manuscript can be accepted.

after correspondence with the editor, I have taken a second look:

I have read the author response again related to the “eximer formation” points raised by reviewer 3. Please note that I am not an expert in photophysics, but:

The answers by the authors seems solid enough.

Their argument that NMR is indication of aggregates is correct, but not necessarily eximer.

They have added extra details in the spectroscopy, and the discussion about eximer is not critical for the main points of the manuscript, and are in general uncontroversial e.g. it is normal that this class of molecules can make eximers.

All together, I think that it would be ok to accept the manuscript in the current form.”

Our response: We sincerely thank the reviewer for his/her valuable time in going through our manuscript and providing the very encouraging comments.

Reviewer #2

“This paper has undergone two rounds of review, and the comments from Reviewer 3 were not really that far apart in terms of mechanism investigation.

My own reading of this paper, is that the design of the compound in this work is interesting by its dual photochromism in absorption and in emission. The authors further explore its ultrawide photoswitchable fluorescence behavior and potential applications.”

Our response: We sincerely thank that the reviewer endorses our novelty.

“I also agree with the comments and suggestions from Reviewer 3 that addressing the mechanism of emission would enhance the quality of this paper. Additionally, I understand the authors' statement about their limitations in investigating the mechanism more thoroughly within a short timeframe due to their focus on material science and limited laboratory equipment.”

Our response: We sincerely thank that the reviewer understands the difficulty/challenge.

“Given the complexity of the system, where spectral signals overlap and obscure each other, it is challenging for me to provide a definitive answer to the editor's query

regarding the evidence supporting the formation of an excimer in the process. I hope that the editor can comprehend this difficulty and exercise their judgment on this issue. Therefore, I would like to defer to the editor's decision.”

Our response: We sincerely thank the reviewer for his/her valuable time in going through our manuscript and providing the very encouraging comments.

The editor's advice:

As a result of the comments from these reviewers, I would ask that either the claims regarding the formation of the excimer are toned down, or that additional evidence is provided to support these claims.”

Our response: Per suggestion, we have modified the related description in our revised manuscript to tone down the claim of excimer formation.

Furthermore, we have carefully checked the manuscript and its Supplementary Information. We have also provided all files according to the Author Checklist. With these changes and point-by-point response to the reviewers' comments and editorial requests, we hope that the revised manuscript is now acceptable for publication.

Your kind consideration of the revised manuscript will be greatly appreciated.